# Evaluating the effects of columnar NO₂ on the accuracy of aerosol optical properties retrievals

Theano Drosoglou[1], Ioannis-Panagiotis Raptis[1,2], Massimo Valeri[3], Stefano Casadio[3], Francesca Barnaba[4], Marcos Herreras-Giralda[5], Anton Lopatin[5], Oleg Dubovik[6], Gabriele Brizzi[3], Fabrizio Niro[7], Monica Campanelli[4], Stelios Kazadzis[8]

[1]Institute for Environmental Research and Sustainable Development, National Observatory of Athens (IERSD/NOA), 15236 Athens, Greece

[2]Laboratory of Climatology and Atmospheric Environment, Sector of Geography and Climatology, Department of Geology and Environment, National and Kapodistrian University of Athens, Athens, GR-15784, Greece

[3]Serco Italia S.p.A., Frascati, Rome, Italy

[4]National Research Council, Institute of Atmospheric Sciences and Climate, CNR- ISAC, Rome, Italy

[5]GRASP SAS, Remote Sensing Developments, 59260 Lezennes, France

[6]Univ. Lille, CNRS, UMR 8518 - LOA - Laboratoire d'Optique Atmosphérique, Lille, France

[7]ESA-ESRIN, Frascati, Rome, Italy

[8]Physicalisch-Meteorologisches Observatorium Davos, World Radiation Center, CH-7260 Davos, Switzerland

*Correspondence to*: Theano Drosoglou (tdroso@noa.gr)

**Abstract.** We aim to evaluate the NO₂ absorption effect in aerosol columnar properties, namely the Aerosol Optical Depth (AOD), Ångström Exponent (AE) and Single Scattering Albedo (SSA), derived from sun-sky radiometers as well as the possible retrieval algorithm improvements by using more accurate characterization of NO₂ optical depth from co-located or satellite-based real-time measurements. For this purpose, we employ multiannual (2017-2022) records of AOD, AE and SSA collected by sun photometers at an urban and a suburban site in the Rome area (Italy) in the framework of both the AERONET and SKYNET networks. The uncertainties introduced in the aerosol retrievals by the NO₂ absorption are investigated using high-frequency observations of total NO₂ derived from co-located Pandora spectroradiometer systems as well as space-borne NO₂ products from the Tropospheric Monitoring Instrument (TROPOMI). For both AERONET and SKYNET, the standard network products were found to systematically overestimate AOD and AE. The average AOD bias found for Rome is relatively low for AERONET (~0.002 at 440 nm and ~0.003 at 380 nm) compared to the retrieval uncertainties, but quite higher for SKYNET (~0.007). On average, an AE bias of ~0.02 and ~0.05 was estimated for AERONET and SKYNET, respectively. In general, the correction seems to be low for areas with low columnar NO₂ concentrations, but it is still useful for low AODs (< 0.3), where the majority of observations are found, especially under high NO₂ pollution events. For the cases of relatively high NO₂ levels (> ~0.7 DU), the mean AOD bias was found within the range 0.009 – 0.012 for AERONET, depending on wavelength and location, and about 0.018 for SKYNET. The analysis does not reveal any significant impact of the NO₂ correction on the derived aerosol temporal trends for the very limited data sets used in this study. However, the effect is expected to become more evident for trends derived from larger data sets as well as in the case of an important NO₂ trend. In addition, the comparisons of the NO₂-modified ground-based AOD data with satellite retrievals from the Deep Blue (DB)

algorithm of the NASA Moderate Resolution Imaging Spectroradiometer (MODIS) resulted in a slight improvement in the agreement of about 0.003 and 0.006 for AERONET and SKYNET, respectively. Finally, the uncertainty in assumptions on $NO_2$ seems to have a non-negligible impact on the retrieved values of SSA at 440 nm leading to an average positive bias of about 0.02 (2 %) in both locations for high $NO_2$ loadings (> 0.7 DU).

## 1 Introduction

Atmospheric particles possess both direct and indirect effects on Earth's radiation budget and climate (IPCC, 2021). Direct radiative forcing arises from aerosols interaction with solar radiation through absorption and scattering processes (Hobbs, 1993). As an indirect impact, aerosols play an important role in cloud formation and properties by acting as cloud condensation nuclei on which water vapour condenses and by influencing the cloud albedo and lifetime (Rosenfeld et al., 2014). Moreover, heterogeneous chemical reactions can take place on the surfaces of atmospheric particles having a crucial effect on atmospheric

chemistry and composition. Examples of such aerosol-driven reactions are those that lead to stratospheric ozone depletion in the polar regions (Solomon et al., 1986). In addition to their footprint on radiative forcing and climate, aerosols adversely affect human health and have been associated with a wide variety of health issues such as respiratory and neurological diseases, cancer, diabetes, cardiovascular diseases and hypertension (e.g., Lelieveld et al., 2015; Molina et al., 2020 and references therein).

The above effects of airborne particulate matter on Earth's climate and human health strongly depend on the intra-annual variations in its loading and properties. The most widely used variable for the estimation of columnar aerosol concentration in the atmosphere is the multi-wavelength Aerosol Optical Depth (AOD). Aerosol optical properties are monitored globally by satellite, e.g., the Moderate Resolution Imaging Spectroradiometer (MODIS) and ground-based networks of sun-photometers like the Aerosol Robotic Network (AERONET, Holben et al., 1998), SKYNET (Nakajima et al., 2020) or the Global

Atmosphere Watch Precision Filter Radiometer (GAW-PFR) network (Kazadzis et al., 2018a). Ground-based remote sensing allows accurate AOD retrievals, i.e. in the order of 0.01 – 0.02 depending on the AOD wavelength (Kazadzis et al., 2018b), which are in fact widely used as a validation reference for satellite or model-based AOD products (e.g., Chu et al ., 2002; Remer  et al., 2005; Green et al., 2009; Levy et al., 2010; Li et al., 2015; Sherman et al., 2016; Gkikas et al., 2021; Di Tomaso et al., 2022) and used as inputs on various modelling initiatives (e.g., Benedetti et al., 2018).

However, AOD retrieval from sun-photometers includes some assumptions in order to take into account all the non-aerosol effects in the retrieval spectral range. In particular, AOD retrievals are sensitive to the assumptions on the concentration of atmospheric trace gases, absorbing in the instrument spectral bands considered, among which are ozone ($O_3$) and nitrogen dioxide ($NO_2$). The exact effect of trace gases in the retrieval at a particular bandwidth depends also on their absorption cross-section. For the case of $NO_2$, as filter radiometers retrieve the AOD in certain wavelength bands, based on their filter

responsivity, such retrievals, especially in the standard wavelengths of 380 and 440 nm (AERONET), have to be corrected for the $NO_2$ optical depth. Currently, some AOD retrievals do not take $NO_2$ optical depth into consideration when deriving AOD

(e.g., SKYNET, Nakajima et al., 2020; GAW-PFR, Kazadzis et al., 2018a) while others use satellite-based climatological $NO_2$ data sets for estimating it (e.g., AERONET, Giles et al., 2019). In the case of the GAW-PFR network, the error introduced in AOD retrievals by $NO_2$ absorption can be assumed to be negligible due to the low $NO_2$ concentrations observed in the GAW remote stations (the annual mean values of $NO_2$ optical depth are in general < 0.001) (Kazadzis et al., 2018a). However, especially over polluted areas, $NO_2$ is characterized by rather short lifetime and high spatiotemporal variations, due to inhomogeneous local emission patterns and photochemical destruction (e.g., Richter et al., 2005; Boersma et al., 2008; Tzortziou et al., 2014, 2015; Drosoglou et al., 2017; Fan et al., 2021). Although the stratospheric component of $NO_2$ is quite stable spatially, the tropospheric $NO_2$ is highly variable in space and time and can bias the calculation of AOD if neglected (Arola and Koskela, 2004; Boersma et al., 2004). Hence, areas with high tropospheric $NO_2$ emission will tend to have greater proclivity for deviating from climatological mean values which might not be representative of the actual $NO_2$ loading and spatial distribution in the atmosphere, introducing potential errors in AOD calculations in those spectral regions with significant $NO_2$ absorption footprint.

Satellite observations with improved spatial and temporal resolution, e.g., the Sentinel-5Precursor Tropospheric Monitoring Instrument (S5P/TROPOMI), models, or co-location with surface-based Pandora instruments from Pandonia Global Network (PGN) spectroradiometers (Cede et al., 2020) measuring total column of $NO_2$ may assist in reducing the uncertainty of the $NO_2$ optical depth contribution in later versions of AOD retrieval algorithms. In the present study, we aim at evaluating if and how much AOD as well as its spectral variability, i.e., the associated Ångström Exponent (AE), and Single Scattering Albedo (SSA) retrievals could be improved by applying a specific correction using synchronous and co-located measurements of total $NO_2$ column from Pandonia network spectroradiometers. To this purpose, we exploit the unique configuration of twin observational sites in the Rome area (Italy), where multiannual (2017-2022) records of both multispectral AOD observations and columnar $NO_2$ measurements are available both in the city centre and in a suburban location. High-frequency measurements of total $NO_2$ performed by co-located Pandora spectroradiometer systems were used to evaluate the current uncertainty in the retrievals of aerosol properties. Aerosol retrieval modifications based on Pandora $NO_2$ measurements are proposed for both AERONET and SKYNET. In addition, relatively high spatially resolved $NO_2$ observations from the S5P/TROPOMI satellite sensor were used to demonstrate the possibility of applying the corrections globally. A first attempt to investigate the impact of those corrections on AOD and AE annual trends is also conducted.

## 2 Instrumentation, data and methodology

### 2.1 The target area and relevant observational sites

Rome is the capital and the most populous city of Italy with almost 3 million inhabitants and one of the most densely populated cities in the European Union (ISTAT 2021). It is located about 24 km east of the Tyrrhenian Sea, surrounded by an extensive undulating plain and crossed by the Tiber and Aniene rivers. The city is part of the Lazio administrative region in the central part of the Italian peninsula. The economic activities in the metropolitan area are characterized by the absence of heavy

industrial facilities and are related mainly to the services and high-technology sectors, as well as commercial activities and tourism. The city air quality is strongly affected by local emission sources, such as transportation and domestic heating, but it is also markedly affected by local circulation and mid-to-long range transport events of sea salt, wildfires and Saharan dust (e.g., Ciardini et al., 2012; Gobbi et al., 2013; Barnaba et al., 2017; Valentini et al., 2020; Di Bernardino et al., 2021).

Rome air quality is monitored on a regular basis by standard in situ instrumentation. These measurements are complemented by multi-platform, long-term observations of aerosol and trace gases performed by a variety of ground-based remote sensing instruments such as sun-sky radiometers, Raman and elastic lidars, automated lidar-ceilometers, Pandora, Brewer and DOAS spectrophotometers (e.g., Di Ianni et al., 2018; Iannarelli et al., 2021; Diemoz et al., 2021). In this study, we used remote sensing measurements of columnar $NO_2$ and aerosol properties performed in two stations located in the greater area of Rome. More specifically, observations were obtained from an urban station (APL-SAP hereafter) located at the Atmospheric Physics Laboratory of the Physics Department of 'La Sapienza' University in the city centre (41.90° N, 12.52° E; altitude 75 m a.s.l.) and a suburban site at the southern east edge of the city in the CNR-ISAC Rome Atmospheric Supersite, CIRAS, in Rome-Tor Vergata (41.84° N, 12.65° E; altitude 117 m a.s.l.). These two observational sites along with the rural station of CNR-IIA in Montelibretti contribute to the Boundary-layer Air Quality-analysis Using Network of Instruments (BAQUNIN) supersite (Iannarelli et al., 2021) as well as to several national and international observing networks.

## 2.2 Aerosol data sets

### 2.2.1 AERONET

The Aerosol Robotic Network (AERONET) is a ground-based passive remote sensing aerosol monitoring network initiated by NASA and expanded by several national and international networks and collaborators (Holben et al., 1998). For more than two decades, AERONET has been delivering continuous, long-term data sets of aerosol optical, microphysical and radiative properties to support aerosol studies and the validation of space-borne retrievals. The network uses the Cimel CE318-T Sun-Sky-Lunar multispectral photometers and provides standardization of instrument calibration and data acquisition as well as centralized data processing and distribution. The AERONET public domain database provides retrievals of spectral AOD, inversion products and precipitable water at a global scale (https://aeronet.gsfc.nasa.gov/, last access: 21 October 2022).

In this study, we employed Level 1.5 quality-assured retrievals of AOD at 380, 440, 500, 675 and 870 nm along with AE at 440-870 nm from Version 3 processing algorithm (Giles et al., 2019; Sinyuk et al., 2020). Level 1.5 data are cloud-screened and quality-assured, but final calibration has not been applied to them. However, they represent a good trade-off between quality and readiness, considering that our approach aims to perform a near-real-time improvement on aerosol products. In the standard AERONET AOD retrieval, the $NO_2$ optical depth is estimated from monthly climatological values of total $NO_2$ from the Ozone Monitoring Instrument (OMI/Aura) Level-3 retrievals during the 2004-2013 period at 0.25° by 0.25° spatial resolution and the $NO_2$ absorption coefficients from Burrows et al. (1998). The observations over the CNR-ISAC station used in this work cover the period March 2017 – mid-August 2022, in which synchronous data from the co-located Pandora

instrument are also available. The respective period for APL-SAP is from April 2017 through early September 2022. The aerosol data sets for both locations are presented in Fig. 1. The average AE is $1.23 \pm 0.4$ and $1.31 \pm 0.5$ at APL-SAP and CNR-ISAC, respectively, while the average AOD is about $0.18 \pm 0.1$ at both stations. AOD has a quite marked yearly cycle, with higher AOD values recorded during summer months, i.e., about $0.22 \pm 0.1$ and $0.21 \pm 0.1$ at APL-SAP and CNR-ISAC, respectively. AE is also higher during summer with a mean value of $1.26 \pm 0.4$ for APL-SAP and $1.38 \pm 0.5$ for CNR-ISAC.

### 2.2.2 SKYNET

The SKYNET network, established at the beginning of the 2000s, is a ground-based radiation observation network dedicated to aerosol, cloud and solar radiation interaction researches using the Prede POM sun-sky radiometers (Takamura and Nakajima, 2004; Nakajima et al., 2020). It is based on the collaboration and maintenance by several universities and research institutes around the world. This network imposes the standardization of instrument calibration, data acquisition and data processing and implements two data analysis flows (SR-CEReS & ESR-MRI) mainly based on the SKYRAD.pack, a software package implemented for the POM sky radiometer (e.g., Nakajima et al., 1996) (https://www.skynet-isdc.org/methodology.php, last access: 21 October 2022). In contrast to AERONET AOD retrieval methodologies, no correction for $NO_2$ optical depth is applied in the calculation of SKYNET AOD (e.g., Campanelli et al., 2004; Estellés et al., 2012). Here, we used the ESR-MRI/SUNRAD processor version 0.9 Level 2 AOD at 400, 500, 675, 870, and 1020 nm and AE at 400-1020 nm data sets over APL-SAP from late September 2017 to May 2022, which are open-accessed online (https://www.skynet-isdc.org/data.php, last access: 21 October 2022). The SKYNET time series used in our analysis is also illustrated in Fig. 1. The calculated mean AOD and AE are $0.18 \pm 0.1$ and $1.23 \pm 0.4$, respectively. These values are similar to AERONET APL-SAP averages mentioned in Sect. 2.2.1, though they correspond to slightly different wavelengths. SKYNET also reports higher values on average during summer, i.e., $0.22 \pm 0.1$ and $1.38 \pm 0.5$ for AOD and AE, respectively.

### 2.2.3 MODIS Deep Blue data

The Moderate Resolution Imaging Spectroradiometer (MODIS) is a key sensor onboard the NASA Terra and Aqua satellites flying respectively since 2000 and 2002. Terra MODIS (descending node, about 10:30 a.m. UTC) and Aqua MODIS (ascending node, about 1:30 p.m. UTC) are observing the entire Earth's surface every 1 to 2 days, acquiring data in 36 spectral bands ranging in wavelength from 0.4 µm to 14.4 µm, with a spatial resolution of 1 km at nadir (except for a few bands with higher spatial resolution).

Inversion of MODIS observations allows retrievals of several geophysical quantities. Here, we used the aerosol AOD products retrieved using the MODIS Deep Blue (DB) algorithm (Hsu et al., 2004, 2006, 2013). The basic principle of DB algorithms is to utilize the pre-calculated land surface reflectance database in deep blue bands (0.412 µm), in which surface reflectance is relatively lower than those in longer bands. In particular, we used the Collection 6.1 DB AOD products for both Aqua and Terra satellites. More details about the DB algorithm are in Hsu et al. (2013) and references therein. The spatial resolution of this product is 10 km. Wei et al. (2019) highlighted that the DB algorithm is relatively more stable and less affected by changes

in atmospheric and surface conditions with respect to the Dark Target algorithm (Levy et al., 2013), showing better performances in urban areas for slightly polluted cases, such as the area of Rome. They also highlighted that Collection 6.1

AOD products perform better than the previous collections, especially in Europe and North America. The MODIS DB products used in this study are available at the Level-1 and Atmosphere Archive and Distribution System Distributed Active Archive Center (http://ladsweb.nascom.nasa.gov, last access: 21 October 2022).

## 2.3 Total $NO_2$ observations

### 2.3.1 Pandora spectroradiometers

Pandora instruments are compact spectrometers that perform spectral measurements with high temporal resolution of direct solar irradiance and scattered radiance for the retrieval of total and tropospheric column densities of atmospheric trace gases (e.g., $NO_2$, $O_3$ and HCHO) that affect air quality, as well as their near-surface concentrations and vertical profiles (e.g., Herman et al., 2009; Tzortziou et al., 2012, 2015). The total $NO_2$ vertical column data sets used in the present study were obtained from the Pandora spectrometer #115 operating at CNR-ISAC since March 2017 and the Pandora systems #117 and #138 both

deployed at APL-SAP since April 2016 and within the period August 2019 – October 2020, respectively. The above time series have been affected by the COVID-19 lockdown period, February – May 2020 (Campanelli et al., 2021). The monthly averaged values from both stations are presented in Fig. 2 and inter-compared in the scatterplot of Fig. 3. On average, the Pandora total $NO_2$ column over APL-SAP is about 0.07% higher compared to the CNR-ISAC $NO_2$.

Pandora total $NO_2$ column product is derived from the direct-sun measurements in the UV-VIS spectral range 280-530 nm

with an average resolution of 0.6 nm by means of the Blick software and the algorithm implemented there as described by Cede (2021). The data sets employed for this work were obtained with the direct-sun retrieval code "nvs3" and the Blick processor version 1.8. Pandora instruments are part of the Pandonia Global Network (PGN) (Cede et al., 2020) and have been fully characterized following the calibration procedures presented by Müller et al. (2020). The recorded raw spectrally resolved radiation measurements are centrally processed for the retrieval of atmospheric trace gas products, which are all publicly

available online (https://www.pandonia-global-network.org/, last access: 21 October 2022). In the current study, high (flags 0 and 10) and medium (flags 1 and 11) quality data are employed. Information on the quality control of Pandora products can be found in Cede et al. (2021). Pandora $NO_2$ retrievals have been compared and validated with other ground-based and space-borne observations during several field campaigns (e.g., Flynn et al., 2014; Martins et al., 2016; Lamsal et al., 2017; Herman et al., 2018; Kreher et al., 2020). Total $NO_2$ data from the Pandora instrument #117 located at APL-SAP have been compared

with $NO_2$ observations retrieved by the co-located MkIV Brewer spectrophotometer with serial number #067, revealing a correlation coefficient above 0.96 and a negligible absolute median bias of 0.002 DU (Diémoz et al., 2021). According to Herman et al. (2009), the Pandora direct-sun total $NO_2$ has a clear-sky precision of 0.01 DU in the slant column and a nominal estimated accuracy of 0.1 DU in the vertical column. In the same study, a systematic difference of less than 1% was found

between the relative slant columns of Pandora and a MultiFunction Differential Optical Absorption Spectroscopy (MFDOAS) instrument.

As already mentioned in Sect. 2.2.1, AERONET uses climatological values from OMI L3 products for the estimation of $NO_2$ optical depth in AOD retrievals. The corresponding OMI total $NO_2$ ranges between about 0.2 and 0.3 DU, with an average value of $0.26 \pm 0.02$ DU. The time series of the Pandora columnar $NO_2$ differences from the AERONET climatological values for both urban (APL-SAP) and suburban (CNR-ISAC) locations is illustrated in the upper panel of Fig. 4. Pandora $NO_2$ data are time-interpolated to AERONET measurements. The percentage frequency distributions of absolute Pandora-OMI deviation for both locations are also presented (Fig. 4, lower panel). About 89% of the APL-SAP and 87% of the CNR-ISAC data pairs show an OMI climatology systematic underestimation of $NO_2$ (positive deviations in Fig. 4). AERONET aerosol retrievals seem to significantly underestimate the $NO_2$ abundance over urban and suburban locations with an average absolute difference between the actual Pandora measurements and the estimations from satellite climatology of about $0.15 \pm 0.19$ DU ($61.5 \pm 71.5\%$) and $0.16 \pm 0.18$ DU ($61.5 \pm 67.2\%$) for APL-SAP and CNR-ISAC, respectively. This underestimation of the $NO_2$ levels over urban locations, characterized by strong spatial gradients, can be attributed to the fact that OMI climatology cannot capture the temporal and spatial $NO_2$ variability within an urban context (e.g., Drosoglou et al., 2017; Herman et al., 2019). Thus, the derived differences in total $NO_2$ are highly correlated to the Pandora measurements. The majority of PGN – OMI biases lie within 0 - 0.5 DU corresponding to Pandora values lower than 1 DU. More specifically, 90% of the PGN $NO_2$ data over APL-SAP differ within -0.14 DU (-50%) and 0.44 DU (150%) from OMI climatology, while the respective deviation range between -0.14 and 0.51 DU (-50% - 170%) for CNR-ISAC. However, there are quite a few cases (~9.5% and ~8.8% for APL-SAP and CNR-ISAC, respectively) of higher PGN values (< 2 DU) leading to larger deviations (up to ~1.6 DU for APL-SAP and ~1.5 DU for CNR-ISAC).

### 2.3.2 TROPOMI

The Tropospheric Monitoring Instrument (TROPOMI) is a nadir-viewing spectrometer on board Sentinel-5 Precursor (S5P) satellite, launched on 13 October 2017. Since August 2019, TROPOMI has a pixel size of 5.5 km × 3.5 km (the initial resolution was 7 km × 3.5 km). $NO_2$ columns are retrieved using the backscatter solar radiation detected in the spectral window of 405-465 nm (van Geffen et al., 2015) by applying the DOAS technique (Platt, 1994; Platt and Stutz, 2008). The operational TROPOMI $NO_2$ products are generated using the algorithm described by van Geffen et al. (2022), which is an improvement of the $NO_2$ DOMINO algorithm (Boersma et al., 2011) developed by the Royal Netherlands Meteorological Institute (KNMI) for the OMI satellite sensor measurements. Both near-real-time (NRTI) and off-line (OFFL) $NO_2$ data sets are retrieved using the KNMI standard algorithm (Eskes et al., 2022; Eskes and Eichmann, 2022). NRTI data files are available within 3 hours from the measurement, whereas the OFFL data are processed in off-line mode and the respective files are generated a few days after the sensing time (van Geffen et al., 2022).

In this study, the OFFL $NO_2$ retrievals are employed, which are the main S5P/TROPOMI product. The extracted $NO_2$ data set covers the period October 2018 – August 2022 and includes observations obtained from several processor versions; beginning

with version 01.02.00 before March 2019 and going up to version 02.04.00 after July 2022. The total $NO_2$ column was calculated from the sum of the tropospheric and stratospheric components, which is preferred over the TROPOMI total $NO_2$ product for comparisons with ground-based data, because the latter suffers from retrieval uncertainties due to its significant dependence on the ratio of the a-priori tropospheric and stratospheric columnar data (van Geffen et al., 2022). Additionally, the satellite pixels have been filtered to keep only those with QA value > 0.75, corresponding to cloud radiance fraction < 0.5 (Eskes and Eichmann, 2022). The S5P/TROPOMI $NO_2$ products have been downloaded from the Sentinel-5P Pre-Operations Data Hub of the Copernicus Open Access Hub (https://scihub.copernicus.eu/, last access: 21 October 2022).

For visualization purposes, averages of the summed $NO_2$ column re-gridded on a 500m grid are plotted for the greater Rome area (Fig. 5). The data used in Fig. 5 cover the period from 2018 to 2021, excluding the COVID-19 lockdown period (February – May 2020) in order to prevent the average $NO_2$ values from being affected by the low values observed during that period.

## 2.4 AOD and AE corrections for NO₂ absorption

### 2.4.1 AOD retrievals

The methodology to derive AOD (also referred to as $\tau$) from photometric measurements is based on the Lambert-Beer law (Eq. 1), which describes light attenuation by atmospheric components. $I_0(\lambda)$ is the intensity of the incident light and $I(\lambda)$ denotes the radiation intensity after traversing through the atmosphere at a specific wavelength $\lambda$.

$$I(\lambda) = I_0(\lambda) \cdot e^{-(m_\tau(\lambda)\tau(\lambda) + m_R(\lambda)\tau_R(\lambda) + \sum_j m_j(\lambda)\tau_j(\lambda))} , \tag{1}$$

$$\frac{\ln I(\lambda)}{\ln I_0(\lambda)} = -(m_\tau(\lambda)\tau(\lambda) + m_R(\lambda)\tau_R(\lambda) + \sum_j m_j(\lambda)\tau_j(\lambda)) , \tag{2}$$

The quantities $\tau$ and $\tau_R$ describe the optical depth of radiation extinction due to aerosols (Mie scattering) and atmospheric molecules (Rayleigh scattering), whereas $m_\tau$ and $m_R$ are the respective air mass factors. $\sum_j m_j \tau_j$ represents the sum of the extinction due to absorption from atmospheric gases (Eq. 3), this depending on wavelength.

$$\sum_j m_j(\lambda)\tau_j(\lambda) = m_{NO_2}(\lambda)\tau_{NO_2}(\lambda) + m_{O_3}(\lambda)\tau_{O_3}(\lambda) + m_{H_2O}(\lambda)\tau_{H_2O}(\lambda) + \dots , \tag{3}$$

In our study we investigate the effects of using an independent, direct measurement of $\tau_{NO_2}(\lambda)$ rather than the climatological value used in the AERONET inversion in determining the AOD ($\tau$). Thus, by combining Eq. (2) with Eq. (3), assuming that the air mass factor in direct-sun measurements is equal to $sec(\theta)$ for both aerosol and $NO_2$, where $\theta$ is the solar zenith angle, and absorption from all the other gaseous components keeps the same, the difference in AOD due to the different estimation of $NO_2$ optical depth is obtained by Eq. (4):

$$\Delta\tau(\lambda) = \tau_{NO_2 PGN}(\lambda) - \tau_{NO_2 AER}(\lambda) , \tag{4}$$

where $\tau_{NO_2 AER}$ is the $NO_2$ absorption optical depth climatology used by AERONET and $\tau_{NO_2 PGN}$ is the optical depth calculated from Pandora $NO_2$ measurements. The latter is derived using Eq. (5):

$$\tau_{NO_2PGN}(\lambda) = \sigma_{NO_2}(\lambda) \cdot c_{NO_2PGN}, \tag{5}$$

The quantity $\sigma_{NO_2}(\lambda)$ in Eq. (5) refers to the absorption cross-section of $NO_2$ at wavelength $\lambda$ (Burrows et al., 1998) and $c_{NO_2PGN}$ is the total $NO_2$ column from Pandora instrument. The modified AOD values ($\tau_{AER\_mod}$) are obtained from the standard AERONET AOD ($\tau_{AER}$) by applying the following equation:

$$\tau_{AER\_mod}(\lambda) = \tau_{AER}(\lambda) - \left( \left( \sigma_{NO_2}(\lambda) \cdot c_{NO_2PGN} \right) - \tau_{NO_2AER}(\lambda) \right), \tag{6}$$

The same approach was also applied to the SKYNET AOD data. However, since the SKYNET retrievals assume $\tau_{NO_2SKYNET} = 0$, Eq. (4) and (6) are modified as:

$$\Delta\tau(\lambda) = \sigma_{NO_2}(\lambda) \cdot c_{NO_2PGN}, \tag{7}$$

$$\tau_{SKYNET\_mod}(\lambda) = \tau_{SKYNET}(\lambda) - \left( \sigma_{NO_2}(\lambda) \cdot c_{NO_2PGN} \right), \tag{8}$$

where $\tau_{SKYNET}(\lambda)$ denotes the standard SKYNET AOD at spectral channel $\lambda$ and $\tau_{SKYNET\_mod}(\lambda)$ is the modified AOD at wavelength $\lambda$.

### 2.4.2 AE retrievals

The spectral variability of AOD is generally expressed as:

$$\tau = \beta \cdot \lambda^{-\alpha}, \tag{9}$$

$$\ln \tau = \ln \beta - \alpha \cdot \ln \lambda, \tag{10}$$

where $\alpha$ stands for the Ångström Exponent (AE).

The AERONET AE product (Eck et al., 1999) is calculated by applying a least squares regression fit on Eq. (10) using the AOD and wavelength logarithms for each non-polarized wavelength channels in different spectral ranges, i.e. 340-440, 380-500, 440-675, 440-870 and 500-870 nm. The negative slope of this linear fit is the Ångström exponent $\alpha$ (Eq. 11).

$$\alpha = - \frac{N \sum \ln \tau_i \ln \lambda_i - \sum \ln \lambda_i \sum \ln \tau_i}{N \sum (\ln \lambda_i)^2 - (\sum \ln \lambda_i)^2}, \tag{11}$$

Here, we also investigate the impact of using synchronous Pandora total $NO_2$ data in AOD algorithm as described in Sect. 2.4.1 on AE retrievals. For this, the AERONET AE product in the range 440-870 nm was used along with the AOD of non-polarized channels included in this range, i.e., 440, 500, 675 and 870 nm. AE was recalculated based on Eq. (11) using the modified AOD at wavelengths 440 and 500 nm obtained from Eq. (6). For the other channels (675 and 870 nm), in which $NO_2$ absorption is negligible, the standard AOD data from AERONET were employed.

For SKYNET, AE is calculated by applying a least squares regression fit on Eq. (10) using the AOD and wavelength logarithms at all wavelengths 400, 500, 675, 870, and 1020 nm. Again, AOD was recalculated using Eq. (8) only at wavelengths 400 and 500 nm, where the impact of $NO_2$ absorption is significant.

The difference in AE due to the different estimation of $NO_2$ optical depth in AOD retrievals is expressed as:

$$\Delta\alpha(\lambda) = \alpha(\lambda) - \alpha_{mod}(\lambda),$$ (12)

where $\alpha_{mod}(\lambda)$ represents the modified AE data and $\alpha(\lambda)$ denotes the AE standard product from AERONET or SKYNET network.

## 2.5 Trend calculations

In this study we also evaluate the impact of modified AOD and AE retrievals, as described in Sect. 2.4.1 and 2.4.2, on aerosol temporal trends. This is only a first attempt to investigate the possible effect of $NO_2$ absorption on the AOD and AE trends since the data sets used here are quite short for statistically meaningful calculations.

The annual trends in AOD and AE were estimated by applying the weighted least squares fitting technique introduced by Weatherhead et al. (1998), previously adopted in several aerosol trend analysis studies from space and the ground (e.g., Zhang and Reid, 2010; Yoon et al., 2012; Logothetis et al., 2021). The applied linear trend model is based on the following formula:

$$Y_t = \mu + \omega X_t + \varepsilon_t, \qquad t = 1, \dots, T,$$ (13)

where $Y_t$ is the monthly average aerosol property of interest, $\mu$ is a constant term representing the linear fit offset at the start of the time series, $\omega$ stands for the magnitude of the trend per year and $\varepsilon_t$ is the monthly average noise not represented by the linear fit. $X_t = {}^t/_{12}$ is the decimal number of years since the first month of the time series, $t$ is the month index, $T$ denotes the total number of months and ${}^T/_{12}$ is the total number of years in the time series.

In order to account for data variability due to severe aerosol events and cloud disturbance, we introduced a monthly weighting factor $w_t$ in the linear fitting procedure (Eq. 14) (Yoon et al., 2012). This weighting factor is defined as the square root of the number of observations available each month $n_t$ divided by the monthly standard deviation $\sigma_t$ (Eq. 15).

$$\chi^2(\mu, \omega) = \sum_{t=1}^{T}\left(w_t \cdot (Y_t - \mu - \omega X_t)\right)^2,$$ (14)

$$w_t = \frac{\sqrt{n_t}}{\sigma_t},$$ (15)

In order to derive statistically significant monthly mean values, a minimum number of 10 observations in a daily basis was ensured. In addition, qualified monthly averages require the availability of measurements from at least 10 days per month. Data were filtered based on the above criteria and days and/or months that did not fulfil them were excluded from the data sample used in the trend calculations. It should be noted that the data sets employed in this study are quite short for statistically

meaningful aerosol trend analysis. However, this is a first attempt to investigate the impact of modified AOD and AE calculations on the derived temporal trends.

## 2.6 GRASP algorithm

The Generalized Retrieval of Atmosphere and Surface Properties (GRASP) (Dubovik et al., 2021) is a state-of-the-art inversion algorithm based on a statistically optimized multi-term Least Square Method (LSM) proposed by Dubovik et al. (2004). GRASP has been applied to numerous applications covering a vast variety of instruments and, which is more interesting, to very different combinations between them. Among the different applications of GRASP, it is possible to find: GRASP/POLDER-3 (Chen et al. 2020), GRASP/AOD (Torres et al., 2017), OLCI/GRASP (Chen et al., 2022), the combination of active lidar measurements and ground-based radiometry (Lopatin et al., 2013; 2021; Román et al., 2018; Herreras et al., 2019), the retrieval of all-sky cameras (Román et al., 2017; 2022) or for example applications to in situ measurements including polar nephelometers (Espinosa et al., 2017; 2019; Schuster et al., 2019).

GRASP scientific core was borne from the heritage of the AERONET inversion algorithm (Dubovik and King, 2000; Dubovik et al., 2000; 2004; King and Dubovik, 2013). At the same time, as discussed above and by Dubovik et al. (2011; 2021), the possibilities of GRASP have been extended due to the totally generalized nature of the inversion module and the continuous developments of the forward model.

For this study, GRASP has been used to mimic AERONET standard retrieval in order to understand the effects of the $NO_2$ concentration on the retrieved SSA at 440 nm. In this case, two different approaches were followed for the GRASP algorithm. First of all, GRASP has been used as close as possible to the standard AERONET retrieval, which means that the input measurements of the algorithm are the Total Optical Depth (TOD) and the almucantar sky measurement routine at 440, 675, 870 and 1020 nm. In the first approach (GRASP/AERONET $NO_2$ hereafter), the $NO_2$ absorption is taken into account using OMI climatology, exactly as in AERONET. On the other hand, GRASP flexibility allows the use of different assumptions of the gaseous properties. Therefore, in addition to the standard approach, the aerosol retrieval has been done also using the total columnar $NO_2$ concentrations provided by the Pandora spectrometers co-located with AERONET instruments at the two stations selected for this study. This methodology will be hereafter referred to as GRASP/Pandora $NO_2$. Thus, in addition to the standard AERONET retrieval products, GRASP has provided aerosol retrieval using these more accurate $NO_2$ concentrations. The $NO_2$ absorption features were calculated more precisely from those concentrations by using a K-Distribution approach; the "kbin" code (Doppler et al., 2014a; 2014b) to speed up the calculations.

## 3 Results and discussion

### 3.1 Differences in AOD and AE retrievals using Pandora $NO_2$ data

The differences in AOD ($\Delta\tau$) at 440 nm and, thus, of its spectral variability through the AE ($\Delta\alpha$ at 440-870 nm) correcting for measured $NO_2$ effects with respect to the standard AERONET retrievals are illustrated in Fig. 6 for both the Rome CNR-ISAC

and APL-SAP stations. The frequency distributions of AOD, $\Delta\tau$ and $\Delta\alpha$ are also included in Fig. 6. $\Delta\tau$ is defined as the standard minus the modified AOD ($\tau_{AER} - \tau_{AER\_mod}$, see Eq. 4-6). Similarly, $\Delta\alpha$ is defined as $\alpha_{AER} - \alpha_{AER\_mod}$ (Eq. 12). The derived values are presented versus the AOD at 440 nm and are color-coded with respect to the Pandora $NO_2$ retrievals. The dependency of $\Delta\tau$ on $NO_2$ is quite clear. As expected, higher $\Delta\tau$ absolute values are obtained for higher $NO_2$ concentrations, regardless of the initial measured AOD. Also, the absolute percentage of $\Delta\tau$ with respect to the AOD is higher for lower aerosol loadings, which means that the impact of the $NO_2$ correction is more significant on lower AODs. This fact is also clear from $\Delta\alpha$, which is higher not only for higher $NO_2$, but for lower AOD values as well. Interestingly, based on Fig. 6, the highest Pandora $NO_2$ retrievals (reddish colors) are not associated with the highest AOD values, indicating that in Rome the high AOD loadings are not strictly associated with high $NO_2$ pollution events. In fact, high AODs are frequently related to long-range transport of elevated layers of desert dust, fires plumes or a combination of both (e.g., Barnaba et al., 2011; Gobbi et al., 2019; Campanelli et al., 2021; Andrés Hernandez et al., 2022). Hence, it might be worth to modify aerosol retrievals for high $NO_2$ in those pollution-related events with low to medium AOD levels. More about AOD and aerosol type climatology for the Rome area can be found in Di Ianni et al., (2018) and in Campanelli et al. (2022).

In general, considering the climatological value chosen for Rome in AERONET retrievals, the use of actual, coincident $NO_2$ measurements on the calculations of aerosol properties seems to be still useful for AOD < 0.3, while quite low (less than 10%) for AOD > 0.5 and almost negligible for AOD > 0.8. In most cases AERONET retrievals seem to overestimate AOD and AE. However, there are cases of underestimation, especially in AE retrievals, which seems to be higher for lower AODs. Those underestimations correspond to overestimation of $NO_2$ from satellite monthly climatological values used in AERONET retrievals. The estimated AOD and AE deviations are below 0.01 and 0.1, respectively, for the majority of observations, i.e., about 96 - 98% of occurrences for both CNR-ISAC and APL-SAP (see also distributions in Fig. 6). The average AOD bias is between $0.002 \pm 0.003$ and $0.003 \pm 0.003$ (with the higher values observed at 380nm), while the average AE bias is $\sim 0.02 \pm 0.03$. Overall, the mean AOD bias is low compared to the estimated uncertainties for the standard AERONET product, i.e., 0.01 - 0.02 (with the higher errors observed in the UV) (Sinyuk et al., 2020). However, the mean AOD bias for the cases of high $NO_2$ levels (> $\sim$0.7 DU) is $\sim 0.011 \pm 0.003$ at 440 nm and $\sim 0.012 \pm 0.003$ at 380 nm for APL-SAP and $\sim 0.009 \pm 0.003$ at 440 nm and $\sim 0.010 \pm 0.003$ at 380 nm for CNR-ISAC, which is comparable to the AERONET reported uncertainties. The estimated mean bias of AE retrievals for the cases with high $NO_2$ (> $\sim$0.7 DU) is $\sim 0.08 \pm 0.04$ for both Rome sites. The threshold for $NO_2$ has been selected as the average Pandora $NO_2$ ($\sim$0.4) calculated from the whole data set plus two times the standard deviation.

The results for SKYNET observations are similar (Fig. 7), but only positive $\Delta\tau$ and $\Delta\alpha$ values are derived, indicating overestimation of the aerosol properties, since the $NO_2$ optical depth is not considered in the standard retrieval processes (see Eq. 7-8). $\Delta\tau$ is defined as $\tau_{SKYNET} - \tau_{SKYNET\_mod}$ (see Eq. 7-8) and $\Delta\alpha$ stands for $\alpha_{SKYNET} - \alpha_{SKYNET\_mod}$ (Eq. 12). In addition, the derived deviations in aerosol properties reach higher values compared to AERONET. Especially AE differences extend up to a value of about 0.7, which is more than double compared to AERONET results. Interestingly, these quite large $\Delta\alpha$ values (> 0.3) correspond to relatively low $NO_2$ loadings (< 1.2 DU). The differences observed between the two networks can be partly

attributed to the different wavelength channels used for AOD and AE retrievals. Similarly to AERONET, the derived AOD and AE biases for SKYNET are below 0.01 and 0.1, respectively, for the majority of observations, i.e., about 85% of occurrences for AOD and about 90% for AE (see also distributions in Fig. 7). The overall average AOD bias is ~0.007 ± 0.003, which can be assumed low considering that Nakajima et al. (2020) have estimated a root-mean-square difference (RMSD) of about 0.03 for wavelengths < 500 nm in city areas in AOD comparisons with other networks. However, the mean AOD bias for the cases with high $NO_2$ levels (> ~0.7 DU) is found to be about 0.018 ± 0.003, which is comparable to the RMSD value reported by Nakajima et al. (2020). The overall average AE bias calculated in this study is ~0.05 ± 0.04, whereas the AE bias averaged over the high $NO_2$ cases is about 0.10 ± 0.05.

WMO (2005) states that, when comparing AOD retrieved from sun-photometers, 95% of the AOD differences should lie within ± (0.005 + 0.01/m) of AOD, where m is the optical air mass. The first term of the equation (0.005) represents the maximum tolerance for the uncertainty due to the atmospheric parameters used for the AOD calculation (additional atmospheric trace gas corrections, i.e., Ozone and $NO_2$, and Rayleigh scattering), while the second term (0.01/m) describes the calibration-related relative uncertainties, for which WMO recommends an upper limit of 1 % (e.g., Cuevas et al., 2019; Kazadzis et al., 2018a). Based on the above, although the average deviations found in this study are low compared to the retrieval uncertainties, they cannot be considered negligible, especially the average systematic underestimation of AOD of about 0.007 from SKYNET, having also in mind that there are locations with much higher average $NO_2$ compared to the city of Rome.

The statistics showing mean differences in AOD and AE AERONET and SKYNET retrievals using actual, coincident $NO_2$ measurements are presented in Table 1. AERONET AOD retrievals at 380 nm are also included in the table. In addition, deviations of AOD and AE using daily or monthly averages of $NO_2$ in AERONET and SKYNET observations are also investigated. Table 1 shows that the average deviations of AOD and AE values do not change significantly whether the actual Pandora $NO_2$ measurements or the daily or monthly mean values are used for the retrievals. The percentage differences for AOD lie within the range 1.2 – 1.9% for AERONET, while they are more than doubled (5.3 – 5.7%) for SKYNET. For the standard aerosol products of the latter, $NO_2$ optical depth is not considered. The estimated percentage differences for AE are within 1.2 - 1.7% and 2.6 – 2.9% for AERONET CNR-ISAC and APL-SAP, respectively, and between 7 – 7.9% for SKYNET APL-SAP. It should be noted that the spectral channels used in AERONET retrievals are 380 and 440 nm for AOD and 440-870 nm for AE, whereas SKYNET data refer to 400 nm and 400-1020 nm for AOD and AE, respectively.

### 3.2 AOD and AE retrievals based on TROPOMI $NO_2$ data

Satellite sensors perform measurements globally and provide information on the air quality even over regions that lack ground-based observations. However, as already mentioned for OMI in Sect. 2.3.1, the spatial resolution of the satellite retrievals is limited by the pixel size. Co-located S5P/TROPOMI observations, characterized by improved spatial and temporal resolution compared to previous satellite missions (e.g., OMI), were also employed to investigate whether the ground-based retrievals of aerosol properties could be improved on a global scale. Again, the approach described in Sect. 2.4.1 and 2.4.2 was applied by

replacing the Pandora total $NO_2$ ($c_{NO_2 PGN}$) with corresponding columnar retrievals from TROPOMI. Based on the current satellite footprint (5.5 km × 3.5 km), a radius of 5 km around each ground-based station was selected for the spatial co-location. The TROPOMI $NO_2$ data were time-interpolated to AERONET and SKYNET measurements. Despite the improved spatial resolution of TROPOMI, the $NO_2$ corrections using TROPOMI data are expected to be less accurate than those performed with the Pandora product. For example, Lambert et al. (2021) showed a bias between TROPOMI and Pandora total $NO_2$ column ranging from -23% over polluted stations to +4.1% over clean areas with a median bias of -7.1%, in the frame of the standard validation process of TROPOMI Level 2 $NO_2$ products. Other studies have concluded similar results. For example, Zhao et al. (2020) showed a negative bias for the standard TROPOMI total $NO_2$ product in the range 23 - 28% over urban and suburban environments and a positive bias of 8 - 11% at a rural site, while Park et al. (2022) showed 26 - 29% negative bias and $R^2$ within 0.73-0.76 over the Seoul Metropolitan Area in Korea.

The statistical metrics of the averaged deviations of the modified AERONET and SKYNET AOD and AE retrievals using actual, co-located TROPOMI $NO_2$ measurements from the network standard products are presented in Table 2. Similarly to Sect. 3.1 and Table 1, deviations of AOD and AE retrievals derived by employing daily or monthly mean TROPOMI total $NO_2$ were also investigated. The average deviations of AOD and AE values do not change significantly whether the actual TROPOMI $NO_2$ measurements or the daily mean values are used for the retrievals. This behaviour is expected considering that TROPOMI overpasses occur once or twice per day and, hence, they do not capture daily variations of $NO_2$. In the case of monthly averaged TROPOMI $NO_2$ data, the estimated differences between the standard and modified aerosol products drop notably for AERONET. However, there are still differences compared to OMI $NO_2$ climatology due to the improved spatial resolution of the TROPOMI pixel. The average AOD bias is ~0.001 ± 0.001 (with the higher values observed at 380nm), while the average AE bias is ~0.01 ± 0.01 for both AERONET stations. For the cases of high $NO_2$ levels (> ~0.7 DU), the mean AOD bias is ~0.004 ± 0.001 at 440 nm and ~0.005 ± 0.002 at 380 nm for APL-SAP and ~0.003 ± 0.001 at both 440 nm and 380 nm for CNR-ISAC. The estimated mean bias of AE retrievals for the cases with high $NO_2$ (> ~0.7 DU) is ~0.05 ± 0.04 and ~0.02 ± 0.01 for APL-SAP and CNR-ISAC, respectively. In the case of SKYNET, the overall average AOD bias is ~0.005 ± 0.002 for AOD and ~0.04 ± 0.03 for AE. For the high $NO_2$ cases, a mean AOD bias of about 0.011 ± 0.002 and an average AE bias of ~0.07 ± 0.04 were calculated. Interestingly, the deviations of SKYNET retrievals using monthly TROPOMI data are very similar to those derived using the actual overpasses or daily averaged TROPOMI $NO_2$, probably due to the fact that the $NO_2$ optical depth is not included in the standard network AOD retrieval processes.

The percentage differences for AOD lie within the range 0.2 – 0.9% for AERONET and are about 3.8 – 3.9% for SKYNET, which are much lower compared to those derived using Pandora $NO_2$ (see Table 1). The estimated percentage differences for AE are ~0.8 – 0.9% and ~1.6 – 1.7% for AERONET CNR-ISAC and APL-SAP, respectively, and about 4% for SKYNET APL-SAP using actual or daily TROPOMI data. It should be noted again that the spectral channels used in AERONET retrievals are 380 and 440 nm for AOD and 440-870 nm for AE, whereas SKYNET data refer to 400 nm and 400-1020 nm for AOD and AE, respectively.

### 3.3 Case study: Impact of high Pandora $NO_2$ on low AOD

In order to investigate further the impact of high $NO_2$ during pollution events on the retrieval of relatively low levels of AOD, we used measurements performed at APL-SAP on June 25[th] 2020, in the morning of which there was a high $NO_2$ event. In the upper panels of Fig. 8, the total $NO_2$ measured from Pandora during that day is illustrated. For AERONET (left panels of Fig. 8), the satellite climatological values used in the retrieval of standard AOD product and their deviations from Pandora $NO_2$ are also displayed. The standard and $NO_2$-modified AOD and AE data from both AERONET and SKYNET (see also Sect. 2.4.1 and 2.4.2), as well as the magnitude of the respective differences ($\Delta\tau$ and $\Delta\alpha$), are presented in the middle and lower panels of Fig.8.

The differences in AOD and AE retrievals from both networks are significant only within a time span of about 3 hours around the high $NO_2$ event (~7:00-10:00 UT) and can be assumed negligible for the rest of the day when the $NO_2$ levels remain quite low. The median AOD bias for AERONET is about 0.003 with a maximum of about 0.02 at the peak of the event. The median and maximum AE biases are 0.014 and 0.11, respectively. It can be also noted that in the case of SKYNET both AOD (median value of ~0.008 with a maximum of ~0.03) and AE deviations (median and maximum values of ~0.03 and 0.10, respectively) are a bit higher compared to the respective AERONET deviations of synchronous data. This can be mainly attributed to the fact that SKYNET standard AOD retrieval processes do not account for the $NO_2$ absorption and partly explained by the different channels used in the detectors of the two networks.

### 3.4 Impact on AOD and AE trends

In this section, a first attempt is conducted to investigate the effect of the modified AOD and AE retrievals based on the Pandora total $NO_2$ observations on the annual trends of those aerosol properties. The annual trends of AERONET/SKYNET AOD and AE over both APL-SAP and CNR-ISAC sites, calculated by applying the approach described in paragraph 2.5, as well as their uncertainties (standard errors of the regression slope) are presented in Table 3.

It should be noted here that the aerosol data sets from the two networks correspond to slightly different time periods. In addition, there are significant gaps in the time series from CNR-ISAC due to instrumental problems and the COVID-19 lockdown period (February – May 2020) has been excluded from the data analysis. Therefore, the results in Table 3 are mainly intended to highlight how a different $NO_2$ correction may affect the aerosol trends and should be interpreted separately for each individual site. Interpretation of the trend significance for the Rome area is not possible using this short period of time (~5.5 years), considering that the estimated trends are quite small and the uncertainties introduced by linear regression are relatively high.

One aspect shown here is that the difference in the AOD and AE trends for the two data sets (original and $NO_2$-modified) is comparable with the calculated trends. As expected, AE trends with and without $NO_2$ correction show relatively higher differences, as AE is much more sensitive to spectral AOD changes. However, the linear fitting uncertainty on AE is also high.

NO$_2$ effects on AOD trends would be more obvious in the case of a significant NO$_2$ trend during a certain period. A thorough long-term trend analysis is out of the scope of this work and could be the topic for a future study.

**3.5 Impact on the inter-comparison of ground-based and satellite AOD data**

In this section, we have analysed a potential effect of considered NO$_2$ corrections on the agreement of AERONET and
SKYNET AOD products with relevant satellite data. Indeed, it is well known that most satellite retrievals are validated against ground-based measurements of AOD that are considered as a ground-truth. Moreover, most satellite retrieval algorithms are substantially tuned to closely match AERONET observations. For example, all MODIS algorithms, including DB, rely, in one way or another, on AERONET dynamic aerosol models and climatologies of AERONET retrievals. Nonetheless, since MODIS retrievals fundamentally rely on MODIS radiances that are fully independent of AERONET data, some inaccuracies in
assumptions, such as those on NO$_2$ amount, can cause some additional biases between AERONET and MODIS AOD results. To evaluate the effects of the proposed correction, we have compared AERONET and SKYNET AOD products against MODIS DB AOD products at 470 nm for the 2017-2022 period. In the inter-comparison, we considered only MODIS DB AOD products for which the distance between the center of the pixel and the AERONET site location (APL-SAP or CNR-ISAC) does not exceed 5 km. Furthermore, we considered all the AERONET (or SKYNET) AOD data within ± 30 minutes
from the MODIS satellite overpasses. In order to guarantee the quality of the data, we used MODIS DB AOD with QA index ≥ 2, which corresponds to good and very good products (Wei et al., 2019).

The inter-comparison has been performed using MODIS DB AOD at 470 nm. Consequently, we computed the AERONET and SKYNET AOD at 470 nm exploiting the AE. The AERONET AOD at 470 nm was calculated using the standard AERONET AOD at 440 nm and AE at 440-870 nm. Similarly, the SKYNET AOD at 470 nm was computed using the standard
SKYNET AOD at 400 nm and AE at 400-1020 nm. The NO$_2$-modified AERONET and SKYNET AOD at 470 nm were also computed with the same approach and the AOD and AE retrievals that have been modified using the Pandora NO$_2$ data.

We observe a generally satisfactory agreement between the ground-based (both AERONET and SKYNET) and MODIS DB AOD data with a Pearson correlation (r) higher than 0.7. In general, MODIS DB AOD slightly overestimates the AOD observed by the sun-photometers. The bias (calculated as satellite minus sun-photometer AOD) between MODIS DB and the
different ground-based data sets before the correction (upper panels of Fig. 9) varies from -0.009 for SKYNET APL-SAP data (-0.008 considering AERONET) to 0.027 for AERONET CNR-ISAC. AERONET data, available for both sites, highlight a lower agreement for the CNR-ISAC site, with a bias about 3 times larger with respect to the APL-SAP site. The correction introduces a slight change of about 0.003 in the agreement between MODIS DB and AERONET AOD products and of 0.006 between MODIS and SKYNET data (lower panels of Fig. 9). Figure 9 also shows an improvement in the percentage of MODIS
AOD data falling within the expected error (EE) of ± (0.05 + 20 %) (Hsu et al., 2013) for APL-SAP adopting the correction on both AERONET and SKYNET.

In Fig. 10, we show the absolute correction (computed as the difference between original AERONET/SKYNET AOD data at 470 nm and modified ones) as a function of the MODIS DB AOD and the NO$_2$ column retrieved by the Pandora instruments

located at APL-SAP and CNR-ISAC sites (upper panels). As already highlighted, we observe that the correction only depends on the $NO_2$ amount and not on the AOD. Figure 10 also highlights that, although the improvement is relatively low on average, the correction can be larger than 10/15% in many cases.

This inter-comparison exercise demonstrated that the proposed correction slightly improves the agreement between MODIS DB AOD data and AERONET and SKYNET AOD products, even if, on average, it is not statistically significant. Nevertheless, as shown in Fig. 10, the improvement becomes significant when the differences between the $NO_2$ values observed by Pandora and the OMI $NO_2$ climatology are also significant (lower panels of Fig. 10). Furthermore, since the proposed correction depends on the amount of $NO_2$, the improvement is more evident in the correspondence of high values of $NO_2$ (upper panels of Fig. 10), typical of highly polluted areas such as the urban area of Rome (APL-SAP). Also, a slight improvement is also achieved in the suburban area of Rome (CNR-ISAC). Finally, in the case of SKYNET AOD products, the systematic overestimation, due to neglected $NO_2$ extinction in the official retrieval chain, is eliminated.

## 3.6 Impact on SSA

One of the main impacts of accurate characterization of the columnar $NO_2$ concentration is certainly expected on the retrieved values of SSA in spectral ranges coinciding with $NO_2$ absorption. In order to quantify this effect, the sensitivity of AERONET retrieval of SSA at 440 nm has been tested. As previously explained (Sect. 2.6), two different GRASP approaches have been applied to this purpose: the GRASP/AERONET $NO_2$ and the GRASP/Pandora $NO_2$. Despite the close methodological basis between GRASP and AERONET retrievals, the divergence in the development of both algorithms has led to some differences in the retrieved products. Thus, in order to assure that the difference in the retrieved SSA at 440 nm is produced exclusively by the changes in the description of $NO_2$ absorption and to avoid the inclusion of any other sources of discrepancy, GRASP code has been used in both approaches instead of the standard AERONET SSA product.

The comparisons of the SSA at 440 nm obtained with both methodologies for the two stations for the complete data set (not shown) do not show a clear influence of the change of $NO_2$ concentration. High correlations (R > 0.98) and a Mean Bias Error (MBE < 0.002) very close to zero are obtained. The mean $NO_2$ column concentration for the retrievals presented here is 0.4 DU. Thus, in general the analysed improvements are not expected to produce an important change in the retrieved parameters at 440 nm in conditions with relatively low $NO_2$ absorption. However, in the cases where $NO_2$ concentration is elevated compared to the climatological expected range, significant changes in the SSA at 440 nm retrievals can be appreciated. Figure 11 shows the comparisons of the SSA at 440 nm obtained with GRASP following an AERONET-like approach (X-axis) and the approach with the new $NO_2$ concentrations provided by Pandora (Y-axis), filtered for $NO_2$ concentrations higher than 0.7 DU, which corresponds to the average $NO_2$ plus two times the standard deviation. The two stations are correspondingly represented in the left and right panels. As it can be noted, for both stations in conditions of high $NO_2$ concentrations there is a consistent positive bias of ~0.02 (~2 %). However, a high correlation (R > 0.96) and Root Mean Square Errors (RMSE < 0.03) are also observed. Previous studies found SSA retrieval uncertainties in the range of 0.02-0.03 (Eck et al., 2003; Corr et al., 2009; Jethva et al., 2014; Kazadzis et al., 2016), whereas the correction, when high $NO_2$ is recorded, is usually higher.

Thus, it is clear that in conditions of high $NO_2$ concentrations an accurate characterization of this gas is necessary in order to avoid noticeable bias in the affected AERONET channel around 440 nm.

**4 Summary and conclusions**

The retrievals of aerosol properties from sun-photometers may be affected by $NO_2$ absorption in the observed spectral range and, thus, accurate assumptions on $NO_2$ concentrations are highly desirable. Currently, some ground-based aerosol networks, such as SKYNET, do not take $NO_2$ optical depth into consideration in AOD retrieval processes, while others (e.g., AERONET) use satellite-based $NO_2$ climatology for estimating it. However, significant errors could be introduced in the AOD retrievals, especially over urban areas, where $NO_2$ variability can be high and also the occurrence of high $NO_2$ events is more frequent.

Such errors may occur only in the cases where $NO_2$ is not taken into account or the used $NO_2$ climatology underestimates such high-$NO_2$ events.

Actual co-located surface-based $NO_2$ measurements (e.g., from Pandora instruments) or space-borne observations with improved spatial and temporal resolution (e.g., S5P/TROPOMI) may be helpful for reducing the uncertainty of the $NO_2$ optical depth contribution in later versions of AOD retrieval algorithms. In this study, we evaluated the possible improvements of

AOD and AE retrievals by applying a specific correction using synchronous and co-located measurements of the total $NO_2$ column from Pandora spectroradiometers and the TROPOMI satellite sensor. For this purpose, we used multiannual (2017-2022) observations from both AERONET and SKYNET multispectral AOD observations co-located with Pandora instruments and collected over two locations in Rome (Italy) with different anthropic pressure, one in the city centre and the other in a suburban area.

The deviations of the $NO_2$-modified AOD retrievals from the network standard products were investigated. AERONET-used $NO_2$ climatology was found to systematically underestimate Pandora-measured $NO_2$ over both sites. The impact of the correction is higher in the case of SKYNET since the $NO_2$ optical depth is not considered at all in the standard retrieval processes of that network. At the same time, the observed differences in the results between the two networks can be partly explained also by the different channels used for the retrievals. For both AERONET and SKYNET, a low but systematic AOD

overestimation was found. Although in most of the cases the differences are lower than 0.01 for AOD and lower than 0.1 for AE retrievals, the correction can still be useful for lower AODs (< 0.3), where the majority of observations are found, especially under high $NO_2$ pollution events. The mean AOD bias derived for the high $NO_2$ cases (> ~0.7 DU) is ~0.011 ± 0.003 at 440 nm and ~0.012 ± 0.003 at 380 nm for AERONET APL-SAP and ~0.009 ± 0.003 at 440 nm and ~0.010 ± 0.003 at 380 nm for AERONET CNR-ISAC. The mean AE bias for the high $NO_2$ is ~0.08 ± 0.04 for both Rome AERONET sites. In the case of

SKYNET, the mean bias for the cases with high $NO_2$ levels (> ~0.7 DU) is ~0.018 ± 0.003 and ~0.10 ± 0.05 for AOD and AE, respectively. Overall, the average biases in AOD retrievals are systematic but within the reported AOD uncertainties. However, they are important to be reported here, as AOD retrieval uncertainties not linked with instrument calibration (e.g. Rayleigh, ozone and $NO_2$ related optical depths) are considered to have an upper limit of 0.005 as a goal for sun-photometers according

to WMO (WMO, 2005). As expected, the effect of improved $NO_2$ assumption in the retrievals is more evident in both AOD and AE when the actual synchronous ground-based Pandora $NO_2$ measurements are employed compared to the situations when the used correction was based on daily or monthly averaged Pandora data or TROPOMI $NO_2$ retrievals. The use of TROPOMI $NO_2$ data is a demonstration of the possibility for corrections on a global scale. However, the underestimation of $NO_2$ concentrations by TROPOMI compared to Pandora $NO_2$ data for Rome leads to lower AOD corrections.

In addition, a first attempt to evaluate the impact of those corrections on AOD and AE annual trends was conducted. However, the aerosol data sets employed in this trend analysis are quite short for a robust trend analysis. Here, only quantitative comparisons are performed for each individual data set, i.e., corresponding to specific instrument and site, before and after the $NO_2$-based correction. Although the effect of $NO_2$ on the derived trends seems to be insignificant and the linear fit trend calculations introduce uncertainties similar or higher to the $NO_2$ effects on AOD, the more pronounced impact may be expected for trends derived from larger data sets as well as in the case of a significant $NO_2$ trend.

We also investigated the possible effects of the proposed $NO_2$ optical depth correction on the agreement between ground-based and space-borne AOD retrievals. In particular, we compared MODIS DB AOD retrievals at 470 nm with AERONET and SKYNET AOD products. In general, the agreement between ground-based (both AERONET and SKYNET) and MODIS DB AOD is quite good, revealing a correlation coefficient (r) higher than 0.7. The use of Pandora $NO_2$ in the sun-photometers retrievals introduces a slight improvement in absolute values of ~0.003 in the agreement between MODIS DB and AERONET AOD and an improvement of ~0.006 between MODIS and SKYNET observations. Although the impact on the comparisons between space-borne and ground-based observations of AOD is quite small, it can be quite useful for eliminating or decreasing possible biases in the inter-comparisons of satellite and ground-based data in situations with $NO_2$ concentrations typical for highly polluted areas.

Finally, we investigated the impact of using a precise characterization of the total $NO_2$ concentration on the SSA retrieval at 440 nm from AERONET measurements. For this, the GRASP algorithm was used to evaluate the effect of $NO_2$ correction on AERONET aerosol retrievals obtained by inverting TOD and almucantar radiances at 440, 675, 870 and 1020 nm. GRASP aerosol retrieval using the actual total $NO_2$ concentration provided by the co-located Pandora, over both stations selected for this study, were compared with GRASP retrievals mimicking AERONET operational retrievals. The results showed that, in general, the effect in the retrieved parameters at 440 nm under low $NO_2$ absorption conditions was not significant. At the same time, for the cases with high $NO_2$ loadings (> 0.7 DU), important changes in the retrieved SSA were observed, with an average positive bias of 0.02 (2 %) for both locations.

In general, the effect of $NO_2$ absorption can be relatively important in the retrievals of aerosol properties, especially AE, AOD and SSA at 440 nm and 380nm, when $NO_2$ is not included in the retrieval algorithms or in cases where $NO_2$ absorption is significantly higher than the $NO_2$ climatology used. If $NO_2$ absorption is accounted from climatological data, the accuracy of such approach may not be sufficient at locations where $NO_2$ has high diurnal variability during high $NO_2$ concentration episodes that cannot be captured by the satellite climatology. In such situations, the use of accurate co-located $NO_2$ observations, e.g., by Pandora instruments, is highly desirable. Thus, based on the results of this study, the effect of $NO_2$

correction could be considered relatively small for large fraction of the observations, nonetheless the correction has certainly contributed towards lowering the uncertainty of AOD and, especially, aerosol SSA provided by sun-photometers.

In future studies, the effect of $NO_2$ correction on Absorption Ångström Exponent (AAE) could be explored. AAE is an aerosol optical property that describes the absorption variation with respect to wavelength and is significantly influenced by particle size, shape, and chemical composition used for aerosol characterization and apportionment studies (e.g., Schuster et al., 2006). Since AAE is a function of spectral AOD and SSA, the $NO_2$ correction for certain AOD wavelengths and SSAs, shown in this study, is expected to impact the AAE calculations towards lower values (as the $NO_2$-corrected AOD is systematically lower,

and the corrected SSA is higher).

Finally, the improved technology including real-time $NO_2$ monitoring (e.g., the Pandonia network), real-time satellite-based products at high spatial resolution (e.g., TROPOMI) and the foreseen more precise $NO_2$ products (e.g., from Sentinel 4) tend to positively contribute towards improving retrieved aerosol properties in the spectral range ($\sim$380 – 440 nm) affected by $NO_2$ absorption.

*Data availability.* The AOD and AE products from the Cimel sunphotometer measurements as well as the $NO_2$ optical depth used in the retrievals are available in the AERONET data server (http://aeronet.gsfc.nasa.gov/). The SKYNET AOD and AE data sets were downloaded from the international SKYNET data center (https://www.skynet-isdc.org/data.php). The Pandora total $NO_2$ columns are available in the Pandonia Global Network website (https://www.pandonia-global-network.org/). The

620 S5P/TROPOMI $NO_2$ products were obtained from the Sentinel-5P Pre-Operations Data Hub of the Copernicus Open Access Hub (https://scihub.copernicus.eu/). The MODIS DB products are available at the Level-1 and Atmosphere Archive and Distribution System Distributed Active Archive Center (http://ladsweb.nascom.nasa.gov). The SSA retrievals from the Cimel almucantar measurements can be accessed by contacting the author.

*Author contribution.* The manuscript was prepared by TD, I-PR, and MV. TD and I-PR developed and implemented the correction algorithm for AOD and AE retrievals and conducted the trend analysis. MV and SC conducted the inter-comparison of ground-based AOD with MODIS DB products and performed the S5P/TROPOMI data extraction and visualization. MH-G and AL developed the SSA retrieval algorithm and conducted the analysis on SSA results. SC, FB and MC supervised the maintenance and operation of ground-based instruments as well as the acquisition and curation of the respective data sets. OD

contributed in the discussions on the SSA analysis and the inter-comparisons with MODIS DB. I-PR, SK, GB and FN supervised the investigation and contributed towards methodological ideas and their presentation. All authors reviewed and edited the manuscript.

*Competing interests.* The authors declare that they have no conflict of interest.

*Acknowledgements*. The work has been supported by the European Space Agency (ESA) in the frame of the Instrument Data quality Evaluation and Assessment Service - Quality Assurance for Earth Observation (IDEAS-QA4EO) project. OD and SK acknowledge the European Metrology Program for Innovation and Research (EMPIR) within the joint research project EMPIR MAPP: "Metrology for aerosol optical properties". The EMPIR is jointly funded by the EMPIR participating countries within EURAMET and the European Union. SK would like to acknowledge the ACTRIS Switzerland project funded by the Swiss State Secretariat for Education Research and Innovation.

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

**Table 1.** Deviation of Pandora total NO$_2$ column from satellite climatology used for AERONET retrievals and differences in modified AERONET and SKYNET AOD and AE from the standard products over CNR-ISAC and APL-SAP calculated using actual Pandora total NO$_2$ observations, as well as daily and monthly averaged values of NO$_2$. Note that the spectral channels used in AERONET retrievals are 380 and 440nm for AOD and 440-870nm for AE, whereas for SKYNET the wavelength channels are 400nm and 400-1020nm for AOD and AE, respectively.

| | | PGN NO$_2$ Actual Measurements | | | | | PGN NO$_2$ Daily Mean | | | | | PGN NO$_2$ Monthly Mean | | | | |
|---|---|---|---|---|---|---|---|---|---|---|---|---|---|---|---|---|
| | | AERONET CNR-ISAC | | AERONET APL-SAP | | SKYNET APL-SAP | AERONET CNR-ISAC | | AERONET APL-SAP | | SKYNET APL-SAP | AERONET CNR-ISAC | | AERONET APL-SAP | | SKYNET APL-SAP |
| **NO$_2$** | Channel [nm] | 380 | 440 | 380 | 440 | 400 | 380 | 440 | 380 | 440 | 400 | 380 | 440 | 380 | 440 | 400 |
| | % Mean Deviation | 61.2 | 64.8 | 63.5 | 64.5 | - | 53.2 | 57.2 | 65.3 | 66.4 | - | 45.4 | 49.6 | 59.4 | 60.7 | - |
| | Mean Deviation [DU] | 0.163 | 0.168 | 0.162 | 0.163 | - | 0.141 | 0.147 | 0.167 | 0.169 | - | 0.120 | 0.127 | 0.151 | 0.153 | - |
| | STD [DU] | 0.170 | 0.171 | 0.182 | 0.182 | - | 0.099 | 0.101 | 0.118 | 0.119 | - | 0.033 | 0.035 | 0.062 | 0.062 | - |
| | Minimum Deviation [DU] | $1.3\times10^{-5}$ | $0.3\times10^{-5}$ | $2.83\times10^{-6}$ | $0.6\times10^{-6}$ | - | $0.6\times10^{-5}$ | $0.7\times10^{-5}$ | $0.1\times10^{-5}$ | $0.5\times10^{-5}$ | - | $0.6\times10^{-6}$ | 0.004 | 0.010 | 0.012 | - |
| | Maximum Deviation [DU] | 2.066 | 2.080 | 2.406 | 2.410 | - | 0.803 | 0.815 | 0.773 | 0.777 | - | 0.297 | 0.311 | 0.291 | 0.293 | - |
| **AOD** | Channel [nm] | 380 | 440 | 380 | 440 | 400 | 380 | 440 | 380 | 440 | 400 | 380 | 440 | 380 | 440 | 400 |
| | % Mean Deviation | 1.5 | 1.7 | 1.7 | 1.7 | 5.3 | 1.3 | 1.5 | 1.9 | 1.9 | 5.6 | 1.2 | 1.3 | 1.8 | 1.8 | 5.7 |
| | Mean Deviation | 0.003 | 0.002 | 0.003 | 0.002 | 0.007 | 0.002 | 0.002 | 0.003 | 0.002 | 0.008 | 0.002 | 0.002 | 0.003 | 0.002 | 0.007 |
| | STD | 0.003 | 0.002 | 0.003 | 0.003 | 0.003 | 0.002 | 0.002 | 0.002 | 0.002 | 0.002 | 0.0005 | 0.0004 | 0.001 | 0.0009 | 0.001 |
| | Minimum Deviation | $0.02\times10^{-6}$ | $0.05\times10^{-6}$ | $0.04\times10^{-6}$ | $0.09\times10^{-7}$ | 0.0022 | $0.01\times10^{-5}$ | $0.01\times10^{-5}$ | $0.02\times10^{-6}$ | $0.07\times10^{-6}$ | 0.0029 | $0.01\times10^{-6}$ | $0.06\times10^{-3}$ | 0.0002 | 0.0002 | 0.0052 |
| | Maximum Deviation | 0.034 | 0.030 | 0.040 | 0.035 | 0.043 | 0.013 | 0.012 | 0.013 | 0.011 | 0.018 | 0.005 | 0.004 | 0.005 | 0.004 | 0.010 |
| **AE** | Spectral Range [nm] | 440-870 | | 440-870 | | 400-1020 | 440-870 | | 440-870 | | 400-1020 | 440-870 | | 440-870 | | 400-1020 |
| | % Mean Deviation | 1.7 | | 2.6 | | 7.0 | 1.4 | | 2.7 | | 7.6 | 1.2 | | 2.9 | | 7.9 |
| | Mean Deviation | 0.019 | | 0.021 | | 0.053 | 0.016 | | 0.022 | | 0.057 | 0.012 | | 0.021 | | 0.058 |
| | STD | 0.027 | | 0.026 | | 0.036 | 0.019 | | 0.023 | | 0.041 | 0.011 | | 0.017 | | 0.044 |
| | Minimum Deviation | $0.02\times10^{-6}$ | | $0.01\times10^{-6}$ | | 0.002 | $0.08\times10^{-5}$ | | $0.01\times10^{-4}$ | | 0.002 | $0.12\times10^{-5}$ | | $0.04\times10^{-5}$ | | 0.003 |
| | Maximum Deviation | 0.309 | | 0.291 | | 0.701 | 0.215 | | 0.322 | | 0.621 | 0.139 | | 0.248 | | 0.640 |

**Table 2. Similar to Table 1, using TROPOMI measurements instead of Pandora total NO$_2$ for the estimation of NO$_2$ abundance in AERONET and SKYNET aerosol retrievals.**

| | | TROPOMI NO$_2$ Actual Measurements | | | | | TROPOMI NO$_2$ Daily Mean | | | | | TROPOMI NO$_2$ Monthly Mean | | | | |
|---|---|---|---|---|---|---|---|---|---|---|---|---|---|---|---|---|
| | | AERONET CNR-ISAC | | AERONET APL-SAP | | SKYNET APL-SAP | AERONET CNR-ISAC | | AERONET APL-SAP | | SKYNET APL-SAP | AERONET CNR-ISAC | | AERONET APL-SAP | | SKYNET APL-SAP |
| **NO$_2$** | **Channel [nm]** | 380 | 440 | 380 | 440 | 400 | 380 | 440 | 380 | 440 | 400 | 380 | 440 | 380 | 440 | 400 |
| | % Mean Deviation | 19.5 | 19.8 | 24.1 | 24.3 | - | 18.7 | 19.0 | 23.4 | 23.6 | - | 6.3 | 6.6 | 12.9 | 13.2 | - |
| | Mean Deviation [DU] | 0.053 | 0.053 | 0.064 | 0.064 | - | 0.051 | 0.051 | 0.062 | 0.062 | - | 0.017 | 0.018 | 0.034 | 0.035 | - |
| | STD [DU] | 0.048 | 0.048 | 0.066 | 0.067 | - | 0.046 | 0.046 | 0.064 | 0.064 | - | 0.019 | 0.018 | 0.025 | 0.025 | - |
| | Minimum Deviation [DU] | $6\times10^{-5}$ | $8\times10^{-6}$ | $2\times10^{-5}$ | $2\times10^{-5}$ | - | $3\times10^{-6}$ | $4\times10^{-6}$ | $2\times10^{-6}$ | $3\times10^{-6}$ | - | $2\times10^{-6}$ | $3\times10^{-5}$ | $5\times10^{-8}$ | $9\times10^{-7}$ | - |
| | Maximum Deviation [DU] | 0.408 | 0.422 | 0.565 | 0.567 | - | 0.398 | 0.412 | 0.564 | 0.566 | - | 0.103 | 0.089 | 0.149 | 0.151 | - |
| **AOD** | **Channel [nm]** | 380 | 440 | 380 | 440 | 400 | 380 | 440 | 380 | 440 | 400 | 380 | 440 | 380 | 440 | 400 |
| | % Mean Deviation | 0.5 | 0.6 | 0.9 | 0.8 | 3.8 | 0.5 | 0.5 | 0.8 | 0.8 | 3.8 | 0.2 | 0.2 | 0.5 | 0.5 | 3.9 |
| | Mean Deviation | 0.0009 | 0.0008 | 0.0011 | 0.0009 | 0.0051 | 0.0008 | 0.0007 | 0.0010 | 0.0009 | 0.0051 | 0.0003 | 0.0003 | 0.0006 | 0.0005 | 0.0051 |
| | STD | 0.0008 | 0.0007 | 0.0011 | 0.0010 | 0.0017 | 0.0008 | 0.0007 | 0.0011 | 0.0009 | 0.0017 | 0.0003 | 0.0003 | 0.0004 | 0.0004 | 0.0008 |
| | Minimum Deviation | $1\times10^{-6}$ | $1\times10^{-7}$ | $3\times10^{-7}$ | $4\times10^{-7}$ | 0.0023 | $5\times10^{-8}$ | $6\times10^{-8}$ | $4\times10^{-8}$ | $5\times10^{-8}$ | 0.0024 | $3\times10^{-8}$ | $5\times10^{-7}$ | $7\times10^{-10}$ | $1\times10^{-8}$ | 0.0038 |
| | Maximum Deviation | 0.007 | 0.006 | 0.009 | 0.008 | 0.017 | 0.007 | 0.006 | 0.009 | 0.008 | 0.015 | 0.002 | 0.001 | 0.002 | 0.002 | 0.008 |
| **AE** | **Spectral Range [nm]** | 440-870 | | 440-870 | | 400-1020 | 440-870 | | 440-870 | | 400-1020 | 440-870 | | 440-870 | | 400-1020 |
| | % Mean Deviation | 0.9 | | 1.6 | | 4.0 | 0.8 | | 1.7 | | 4.0 | 0.5 | | 0.6 | | 4.2 |
| | Mean Deviation | 0.009 | | 0.012 | | 0.038 | 0.009 | | 0.011 | | 0.038 | 0.006 | | 0.005 | | 0.039 |
| | STD | 0.011 | | 0.017 | | 0.026 | 0.010 | | 0.016 | | 0.027 | 0.004 | | 0.006 | | 0.027 |
| | Minimum Deviation | $1\times10^{-7}$ | | $1\times10^{-7}$ | | 0.001 | $2\times10^{-7}$ | | $7\times10^{-8}$ | | 0.001 | $1\times10^{-7}$ | | $7\times10^{-8}$ | | 0.002 |
| | Maximum Deviation | 0.116 | | 0.239 | | 0.286 | 0.116 | | 0.238 | | 0.393 | 0.036 | | 0.090 | | 0.252 |

**Table 3. AOD and AE trends and their uncertainties for both standard and modified AERONET and SKYNET products over CNR-ISAC and APL-SAP. Note that the spectral channels used in AERONET retrievals are 440 nm for AOD and 440-870 nm for AE, whereas for SKYNET are 400 nm and 400-1020 nm for AOD and AE, respectively. The trend uncertainties refer to the standard error of the regression slope. The differences are calculated on the absolute trend values.**


| | | AERONET CNR-ISAC | | AERONET APL-SAP | | SKYNET APL-SAP | |
|---|---|---|---|---|---|---|---|
| | | standard | modified | standard | modified | standard | modified |
| | Number of years | 5.7 | 5.7 | 5.7 | 5.7 | 5.4 | 5.4 |
| **AOD** | Trend [/year] | 0.004 | 0.002 | 0.0002 | 0.0005 | 0.002 | 0.002 |
| | % trend [/year] | 2.0 | 1.4 | 0.1 | 0.3 | 0.8 | 1.0 |
| | Uncertainty | 0.005 | 0.005 | 0.005 | 0.005 | 0.006 | 0.006 |
| | Modified – standard | | -0.001 (-33.1%) | | 0.0003 (174.3%) | | 0.0002 (15.8%) |
| **AE** | Trend [/year] | 0.047 | 0.042 | -0.022 | -0.0181 | -0.061 | -0.057 |
| | % trend [/year] | 3.8 | 3.4 | -1.8 | -1.5 | -5.6 | -5.5 |
| | Uncertainty | 0.025 | 0.026 | 0.018 | 0.019 | 0.026 | 0.026 |
| | Modified – standard | | -0.006 (-12.4%) | | -0.004 (-18.1%) | | -0.004 (-6.9%) |


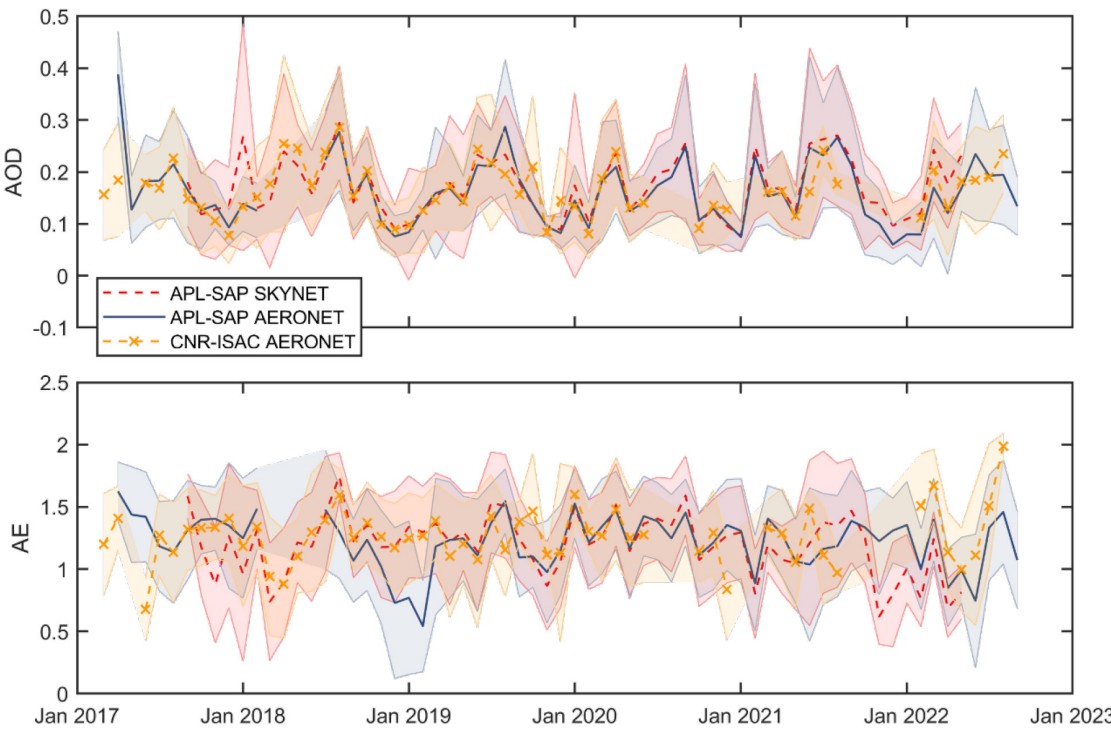

**Figure 1: Time series of monthly averaged AOD (upper panel) and AE (lower panel) measurements over APL-SAP (AERONET and SKYNET) and CNR-ISAC (AERONET). Note that AERONET AOD and AE correspond to the wavelength channels of 440nm and 440-870nm, respectively, whereas SKYNET AOD and AE refer to 400nm and 400-1020nm, respectively. The shaded areas correspond to the monthly 1-sigma standard deviation.**


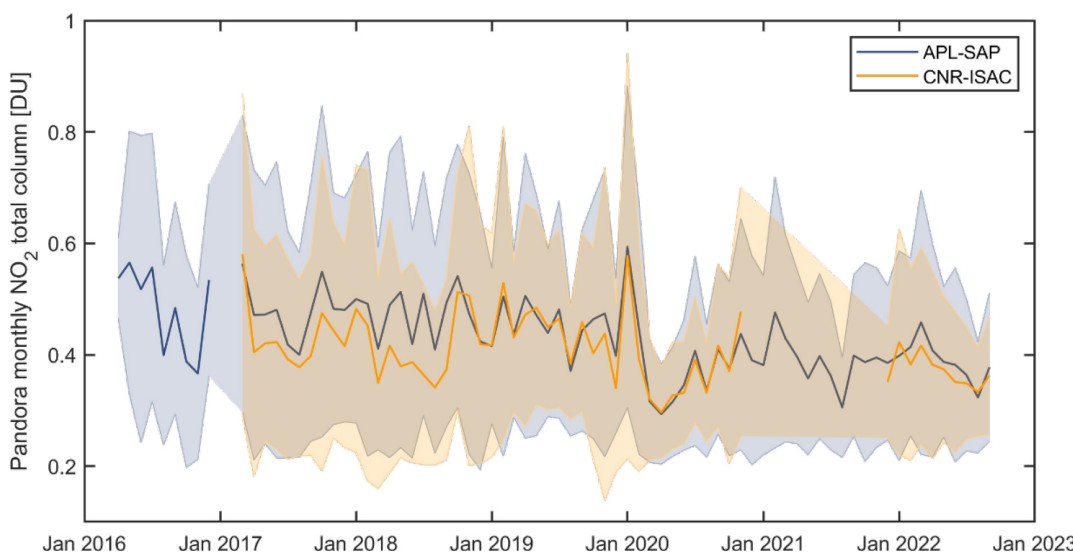

**Figure 2: Time series of monthly NO₂ total column from Pandora instruments over APL-SAP (blue line) and CNR-ISAC (yellow line). The shaded areas correspond to the 1-sigma standard deviation of the monthly averaged values. The NO₂ concentration is clearly affected by the COVID-19 lockdown during February – May 2020.**

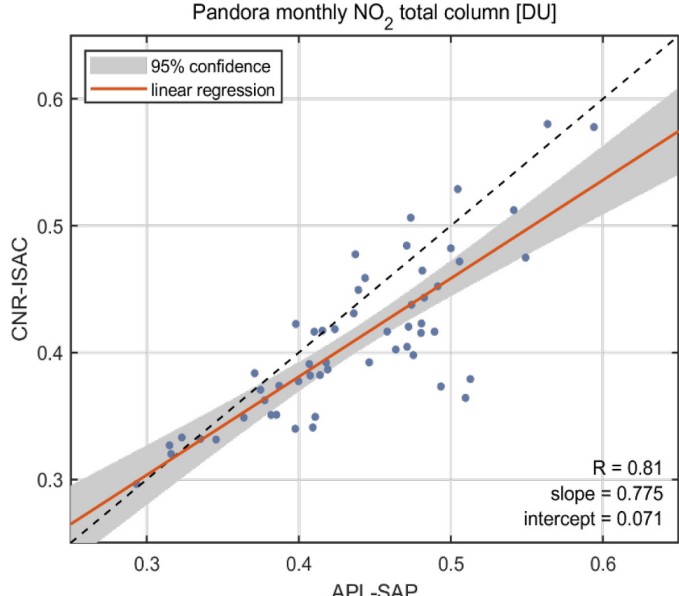


**Figure 3: Monthly NO₂ total column from Pandora over CNR-ISAC against synchronous APL-SAP observations. The grey shaded area corresponds to the 95% confidence interval of the linear regression fit (red line).**


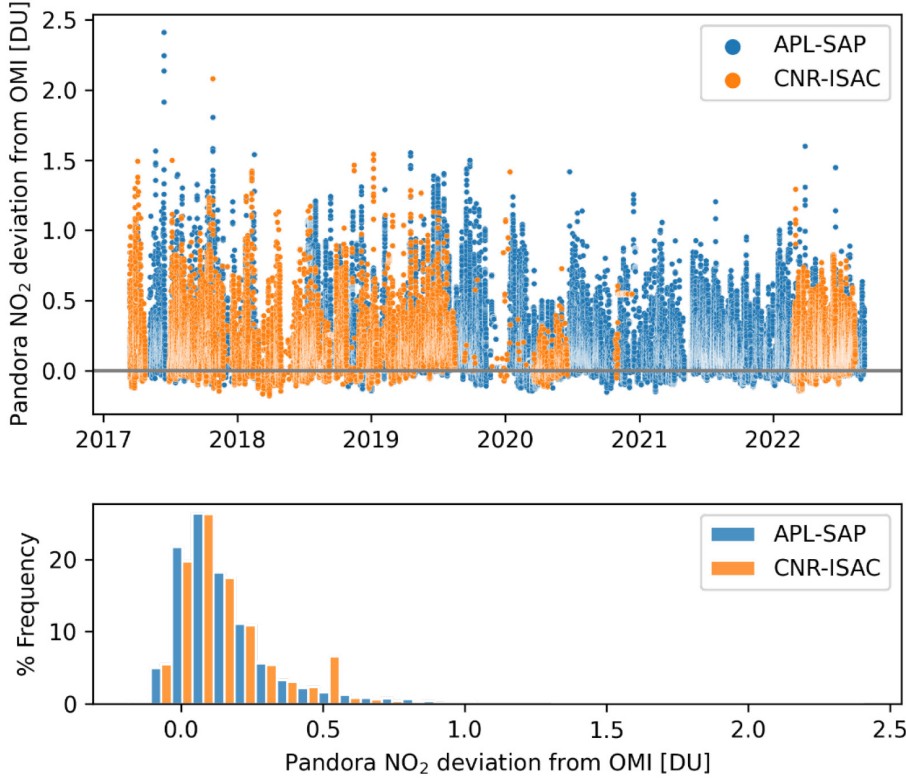

**Figure 4: Time series of the Pandora total NO₂ deviation from AERONET NO₂ climatological values (OMI) for both APL-SAP and CNR-ISAC (upper panel). The corresponding relative frequency distributions of Pandora-OMI deviation for both locations are illustrated in the lower panel.**


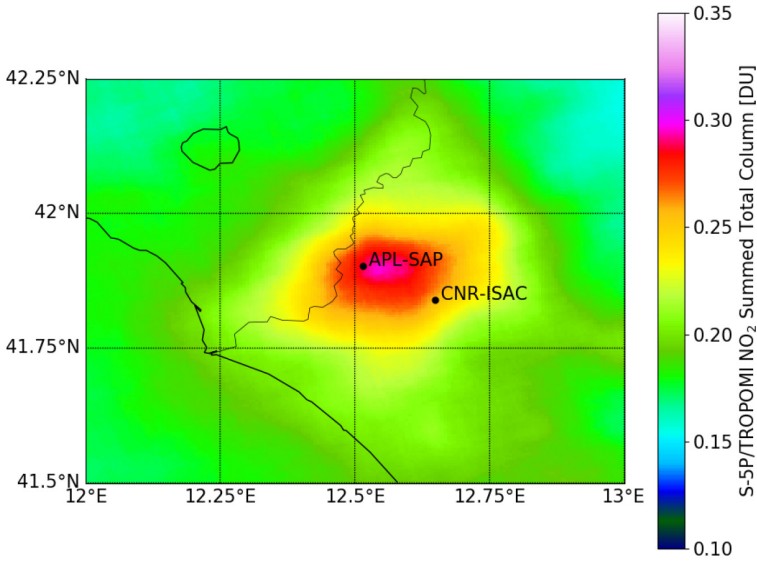

**Figure 5: S5P/TROPOMI summed total NO₂ column averaged for the period 2018-2021, excluding the COVID-19 lockdown period. The data are gridded on a 500m grid. The locations of the two observational sites used in this study are also reported for reference.**


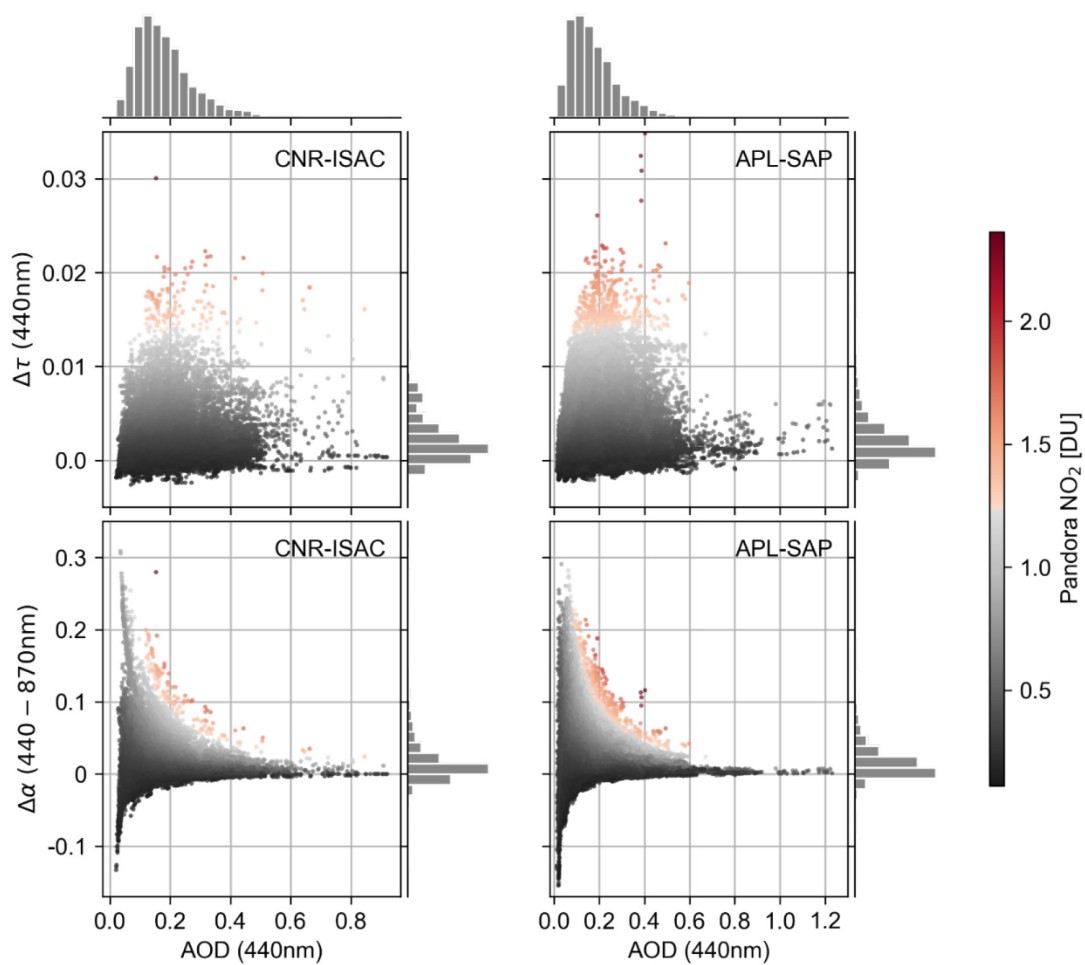

**Figure 6: The differences of modified AERONET AOD at 440 nm (upper panels) and AE at 440-870 nm (lower panels) over CNR-ISAC (left panels) and APL-SAP (right panels) from the standard products illustrated with respect to the standard AERONET AOD measurements at 440 nm and the actual NO₂ observed by Pandora (color scale). The corresponding distributions of all variables are also included.**


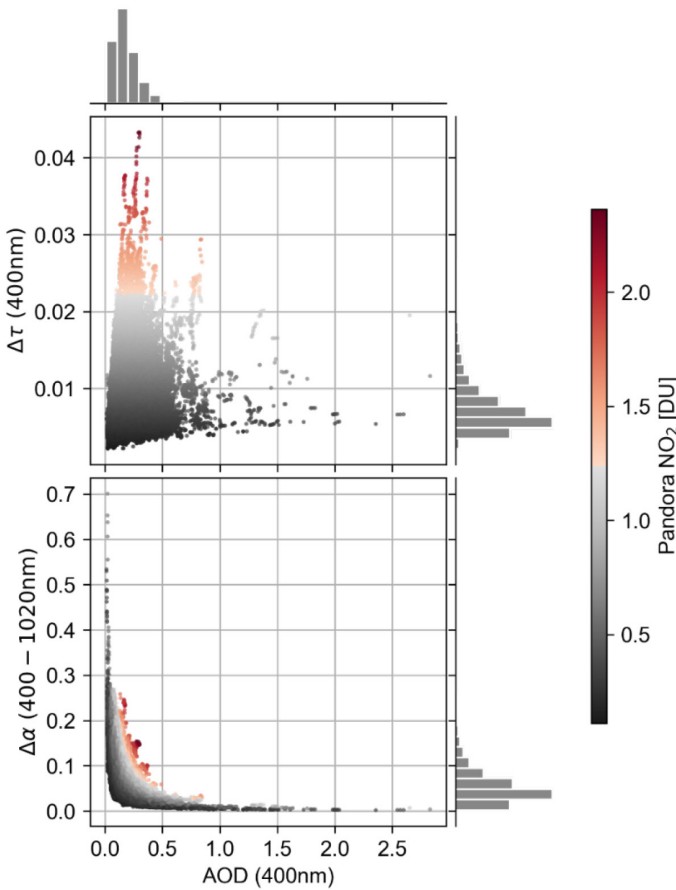

**Figure 7: Similar to Fig. 6, but for SKYNET retrievals over APL-SAP. Note that the spectral channels for the retrievals are different compared to AERONET, i.e. 400nm for AOD and 400-1020nm for AE. Also the axis scales are different.**

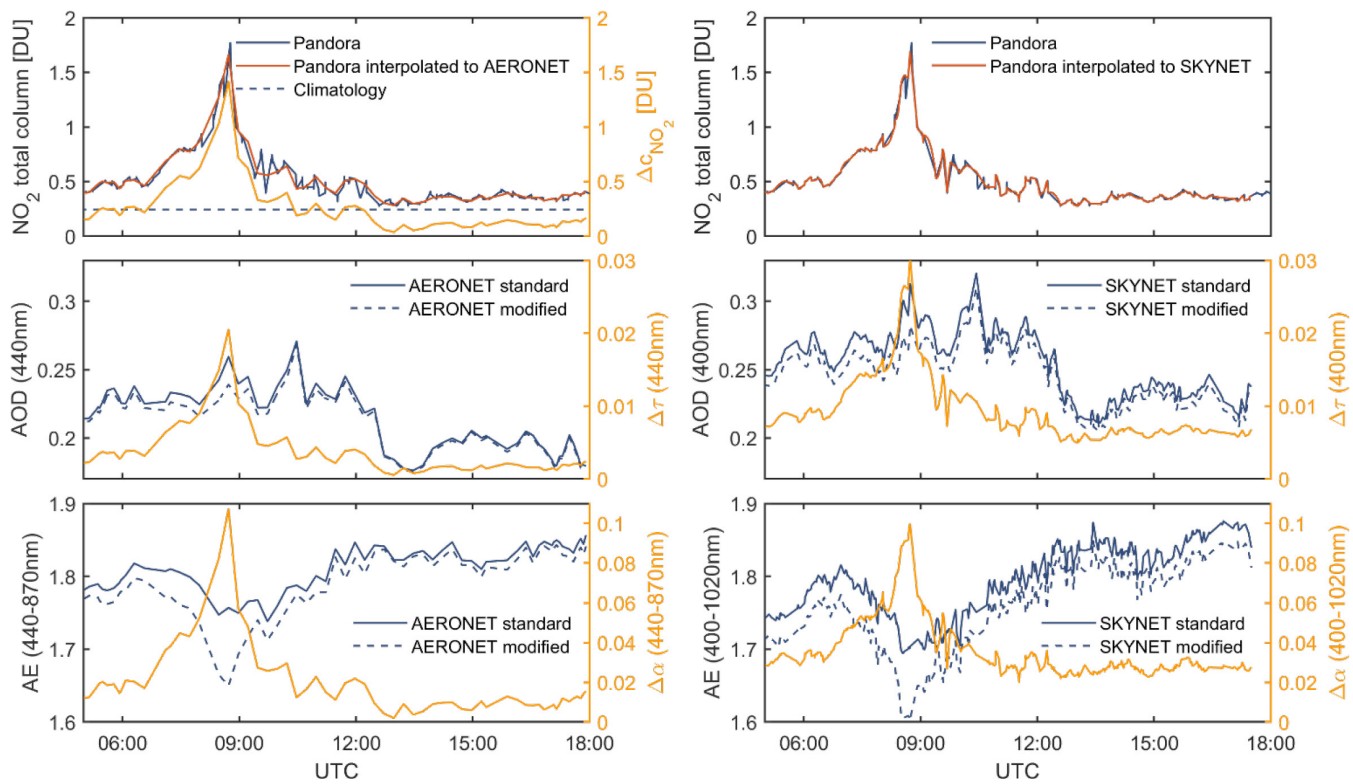

**Figure 8: Case study over APL-SAP for 25th June 2020 for both AERONET (left panels) and SKYNET (right panels). Upper panels: Pandora total NO₂ column and its deviation from climatology. Middle panels: AOD (solid blue line), its improvement using Pandora NO₂ (dashed blue line) and the magnitude of improvement (light orange line and right y-axis). Lower panels: Similar to middle panels, but for AE retrievals. Note that the spectral channels for the retrievals are different for the two networks.**

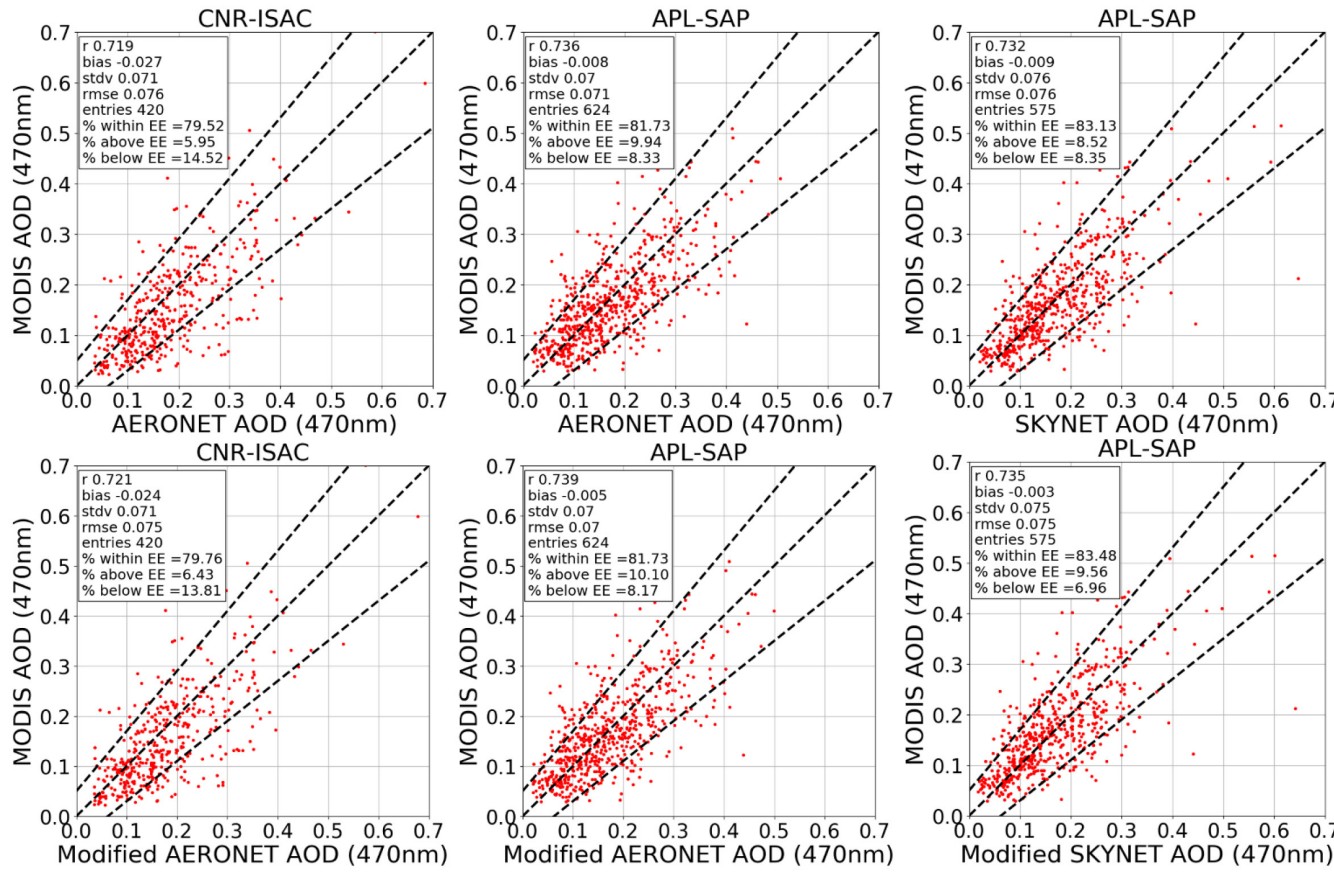


**Figure 9: Inter-comparison of MODIS DB with standard (upper panels) and modified (lower panels) ground-based AOD at 470 nm for CNR- ISAC (left panels) and APL-SAP (middle and right panels) sites. In the left and middle panels, against AERONET AOD products, whereas in the right panels against SKYNET AOD. The y=x lines and MODIS DB EE envelopes ± (0.05 + 20%) are plotted as dashed lines. The inter-comparison was performed considering a maximum distance between the center of the MODIS DB pixel**
**and the site location of 5 km and Δt_max (time between MODIS and AERONET/SKYNET observations) of ±30 minutes.**

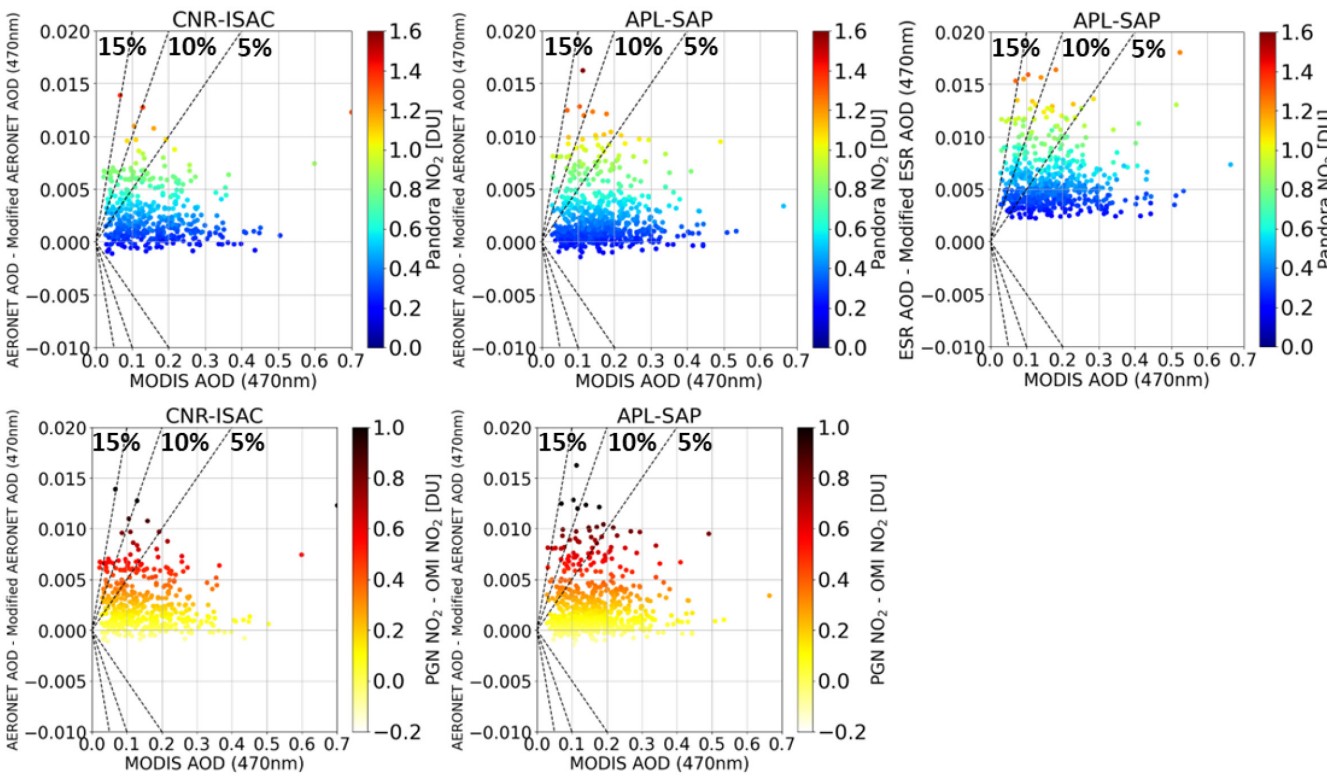

Figure 10: Upper row: Absolute correction as a function of the corresponding MODIS DB AOD data and PGN NO₂ data for CNR-ISAC (left panel) and APL-SAP (middle and right panels) sites. In the left and middle panels the inter-comparison was performed using AERONET AOD products, in the right panel SKYNET AOD was used. The color scale represents the PGN NO₂ retrieved in correspondence of the AERONET/SKYNET AOD products. The analysis was performed considering a maximum distance between the center of the MODIS DB pixel and the site location of 5 km and $\Delta t\_max$ of ±30 minutes. Lower row: As in the upper row, but here the color scale represents the absolute difference between PGN and OMI climatological NO₂ data in correspondence with the AERONET AOD products.

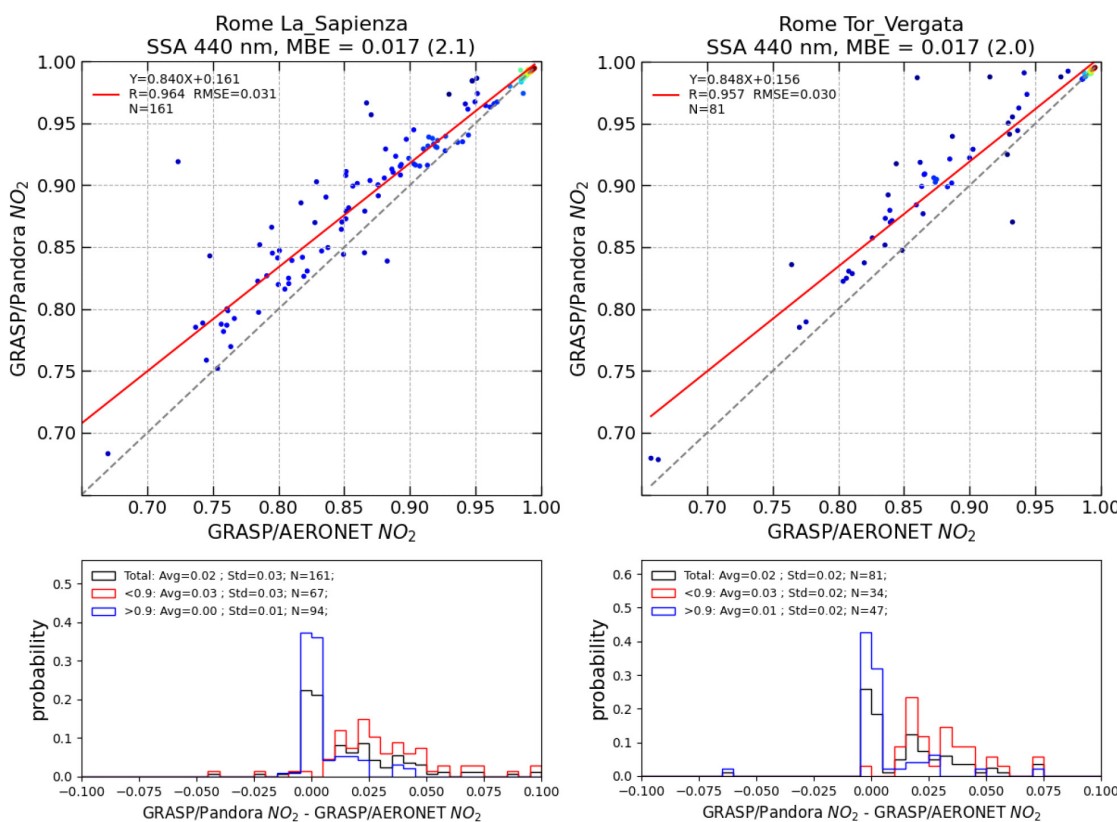


**Figure 11: Comparisons of SSA at 440 nm obtained with GRASP following the standard AERONET procedure (X axis) and a similar approach but precisely accounting for $NO_2$ concentration (Y axis) from the co-located Pandora instruments in two different stations: APL-SAP (right upper panel) from March 2017 to November 2020, and CNR-ISAC (left upper panel) from April 2017 to September 2021. The data has been filtered to show retrievals corresponding to $NO_2$ concentration higher than 0.7 DU. The color of the circles is an indicator of the density of points, i.e., colors closer to red, indicate higher number of points close together. The absolute Mean Bias Error (MBE) (percent in parenthesis), the Root Mean Square Error (RMSE) and the correlation coefficient of the linear fit are also shown in the figure. The probability density functions of the difference between both methodologies (GRASP/Pandora $NO_2$ – GRASP/AERONET $NO_2$) can be found in the lower panels correspondingly for each station. The probability density functions for SSA values higher or lower than 0.9 are also included.**
