# Peer review of "Evaluating the effects of columnar NO2 on the accuracy of aerosol optical properties retrievals"

_Atmospheric Measurement Techniques, 2022_

## Author Comment (AC1)

**Response to anonymous referee #1**

The reviewer comments are given in black, followed by the authors' response in blue. Text copied from the revised manuscript is in blue italic.

Based on a comment from referee #1, Fig. 8 and 9 have been merged and, as a result, the numbering of the following figures has been changed. In addition, the structure of Sect. 2 has been modified in order to include a new subsection with the description and implementation of the GRASP algorithm (Sect. 2.6).

**General comments:**

Overall, this is a well-written paper evaluating an interesting aspect like the effect of the columnar $NO_2$ correction in the accuracy of the Aerosol Optical Depth (AOD), the Angström Exponent (AE) or the Single Scattering Albedo (SSA) using multiannual records (5 years). This paper addresses some important aspects for the scientific community, such as the investigation of the effects of using different $NO_2$ data (no correction, direct retrievals or climatological values), which can impact the aerosol products retrieved with different global aerosol networks (NASA-AERONET, GAW-PFR or SKYNET-Prede). Moreover, $NO_2$ satellite data (TROPOMI) and spectral ground-based data (from Pandonia Global Network, PNG) were used to investigate the possible improvement in aerosol properties retrieved from these three largest ground-based aerosol networks. Trend analysis has been included to understand the impact of the $NO_2$ correction on the derived aerosol products, although the authors make it clear throughout the article that the number of data in the database is insufficient to carry out this type of study.

I consider that this manuscript fits perfectly into the scope of AMT and that the results presented here are relevant. There are only a few minor remarks.

We would like to acknowledge the referee for their helpful and thorough review. We believe that their comments improved the quality of this work.

**Minor comments:**

Abstract: Is uncertainty estimation a goal of this paper? I consider that this work deals with the impact of the columnar $NO_2$ effect rather than evaluating/investigating the uncertainty introduced by this term. Please state.

In principle, AOD uncertainty estimation due to $NO_2$ column used for AOD post processing is the main goal of the paper. The way to treat this theoretically would need an estimation on the used $NO_2$. Therefore, we have tried to demonstrate errors related with the use of climatological $NO_2$ rather than

the actual measurements of $NO_2$. So a first step towards this goal is the evaluation of the satellite climatology used, based on collocated $NO_2$ observations from a state-of-the-art network. The second step is the assessment of these climatology vs measurements difference on AOD retrieval. For instruments not correcting for $NO_2$, it is straightforward that the AOD retrieval bias is directly linked with the $NO_2$ amount as in reality what is defined as AOD is the sum of the aerosol and $NO_2$ optical depth. Finally, as this study was performed on a pilot-test urban area, and can be, thus, considered "local", we tried to assess the use of real-time satellite data for such corrections too.

So in our case the AOD uncertainty is always linked with the $NO_2$ related uncertainty, but here we present an idea of two-instrument synergy towards less uncertain $NO_2$ data, leading to less uncertain AOD data.

Page 2, lines 34-37: The authors are introducing the direct and indirect effects of aerosols. Don't the authors believe that there are more adequate references to introduce these effects? At least one more recent version of the IPCC exists than the one included in this article.

The authors thank the referee for pointing this out. The reference has been updated with a more recent version of the IPCC assessment report:

IPCC, 2021: Climate Change 2021: The Physical Science Basis. Contribution of Working Group I to the Sixth Assessment Report of the Intergovernmental Panel on Climate Change [Masson-Delmotte, V., P. Zhai, A. Pirani, S.L. Connors, C. Péan, S. Berger, N. Caud, Y. Chen, L. Goldfarb, M.I. Gomis, M. Huang, K. Leitzell, E. Lonnoy, J.B.R. Matthews, T.K. Maycock, T. Waterfield, O. Yelekçi, R. Yu, and B. Zhou (eds.)]. Cambridge University Press, Cambridge, United Kingdom and New York, NY, USA, In press, doi:10.1017/9781009157896.

Page 4, line 101: Is "specifically" right in this sentence?

This sentence is used as a more specific description of the two stations mentioned in the previous sentence. Thus, we think that the word "specifically" could be considered okay in this sentence. A comma separator has been added after this word so that the meaning is clearer:

*"In this study, we used remote sensing measurements of columnar $NO_2$ and aerosol properties performed in two stations located in the greater area of Rome. More specifically, observations were obtained from an urban station…"*

Page 4, line 112: The authors write "Cimel" in this sentence but "CIMEL" later in the text. Please homogenize.

The manuscript has been revised accordingly.

Page 4, line 118: Please note that Version 3 Level 1.5 includes data with near-real-time automatic cloud screening and automatic instrument anomaly quality controls while Level 2.0 additionally applies prefield and post-field calibrations. This means that the 1.5 level does not in any way apply the final calibration, so the lack of certainty in the verb "may" does not seem correct.

The word "may" has been removed from the sentence.

Page 4, Fig. 1: The authors present here a time evolution of the AOD and AE observations at APL-SAP and CNR-ISAC. I don't see the point of including such a figure, because these data are not exploited here nor are they mentioned throughout the text.

This figure has been added to provide the reader with an overview of the aerosol data used in this work, and, in particular, to show typical values and seasonal cycles of the aerosol metrics addressed. We think it is a nice introduction to the AOD and AE levels in the two Rome sites. It is also useful as comparison to Fig.2, showing relevant $NO_2$ data, and, thus, highlighting the different seasonal cycles of the variables. For these reasons, we prefer to keep it. Following the suggestion, we included a brief description of the figure in Sections 2.2.1 and 2.2.2 as follows:

*"The aerosol data sets for both locations are presented in Fig. 1. The average AE is 1.23 ± 0.4 and 1.31 ± 0.5 at APL-SAP and CNR-ISAC, respectively, while the average AOD is about 0.18 ± 0.1 at both stations. AOD has a quite marked yearly cycle, with higher AOD values recorded during summer months, i.e., about 0.22 ± 0.1 and 0.21 ± 0.1 at APL-SAP and CNR-ISAC, respectively. AE is also higher during summer with a mean value of 1.26 ± 0.4 for APL-SAP and 1.38 ± 0.5 for CNR-ISAC…*

*…The SKYNET time series used in our analysis is also illustrated in Fig. 1. The calculated mean AOD and AE are 0.18 ± 0.1 and 1.23 ± 0.4, respectively. These values are similar to AERONET APL-SAP averages mentioned in Sect. 2.2.1, though they correspond to slightly different wavelengths. SKYNET also reports higher values on average during summer, i.e., 0.22 ± 0.1 and 1.38 ± 0.5 for AOD and AE, respectively."*

Page 4, last paragraph: The authors have used level 1.5 SSA information from AERONET. However, as stated by Sinyuk et al. (2020), quality-controlled SSA data (level 2.0) should be retrieved for AOD larger than 0.4 and SZA larger than 50°. How the authors have ensured the quality of the SSA information included in this paper? Why the authors have not included AERONET Level 2.0 data in this study?

The levels of uncertainty provided by Sinyuk et al. (2020) clearly state the difference in quality of the different AERONET data levels in the mentioned conditions. However, these restricted conditions imply an extremely reduced amount of available data that makes impossible comparisons with a proper level of statistical significance. Thus, the authors consider that the trade between the amount of data and the loss of accuracy in the retrieved values results beneficially for the final quality of the comparisons. Also note that this methodology has been successfully applied in several publications as for example in Román et al. (2017), Román et al. (2018), Benavent-Oltra et al. (2019), Herreras et al. (2019).

Román, R., Torres, B., Fuertes, D., Cachorro, V. E., Dubovik, O., Toledano, C., ... & Alados-Arboledas, L. (2017). Remote sensing of lunar aureole with a sky camera: Adding information in the nocturnal retrieval of aerosol properties with GRASP code. Remote Sensing of Environment, 196, 238-252.

Román, R., Benavent-Oltra, J. A., Casquero-Vera, J. A., Lopatin, A., Cazorla, A., Lyamani, H., ... & Alados-Arboledas, L. (2018). Retrieval of aerosol profiles combining sunphotometer and ceilometer measurements in GRASP code. Atmospheric Research, 204, 161-177.

Benavent-Oltra, J. A., Román, R., Casquero-Vera, J. A., Pérez-Ramírez, D., Lyamani, H., Ortiz-Amezcua, P., ... & Alados-Arboledas, L. (2019). Different strategies to retrieve aerosol properties at night-time with the GRASP algorithm. Atmospheric Chemistry and Physics, 19(22), 14149-14171.

Herreras, M., Román, R., Cazorla, A., Toledano, C., Lyamani, H., Torres, B., ... & de Frutos, A. M. (2019). Evaluation of retrieved aerosol extinction profiles using as reference the aerosol optical depth differences between various heights. Atmospheric Research, 230, 104625.

Page 6, lines 183-185: The authors introduce here a past comparison between Pandora and Brewer without giving any result of this comparison. This sentence seems dispensable if it does not provide more information about the validity of Pandora $NO_2$ data.

Comparison results have been included in the text as follows:

*"Total $NO_2$ data from the Pandora instrument #117 located at APL-SAP have been compared with $NO_2$ observations retrieved by the co-located MkIV Brewer spectrophotometer with serial number #067, revealing a correlation coefficient above 0.96 and a negligible absolute median bias of 0.002 DU (Diémoz et al., 2021)."*

Page 6, last paragraph: This paragraph explains the $NO_2$ deviation Pandora versus OMI (AERONET) as is displayed in Fig. 4. It is written that, according to Fig. 4 (lower panel), biases (Pandora-OMI, I guess) of 89% and 87% are found. I'm not able to see these results in the lower panel. Later, the authors give another result: Pandora-OMI average differences of 61.5% at both stations. Could you please explain more in detail these different results and where do they come from?

The distribution of Pandora-OMI deviations are shown in the lower panel of Fig. 4. It should be noted that Fig. 4 have been changed, but the lower panel still shows the same distributions as before, without stacked bars. The peaks of the distribution are between 0-0.2 DU and the calculated mean biases (with the respective percentage values) are 0.15 ± 0.19 DU (61.5 ± 71.5%) and 0.16 ± 0.18 DU (61.5 ± 67.2%) for APL-SAP and CNR-ISAC, respectively. The 89% and 87% values are not the estimated biases. These numbers indicate the percentage of the Pandora-OMI data pairs with positive deviations, i.e. for how many cases the Pandora values are higher compared to the respective OMI values. Since the initial text might be misleading, it has been revised.

Page 9, line 281: As mentioned before, the authors acknowledge throughout the text that this database is too short to perform statistically meaningful trend analysis. The question is obvious: why then carry out this analysis?

The purpose of the analysis is not to derive any conclusions on the AOD or AE trends in the two Rome sites. The sole purpose of showing the trend analysis is to quantify how much the trend differs for the

standard and the modified AODs. The importance of this difference can be assessed only quantitatively as aspects such as the trend analysis regression uncertainty and the trend "levels" compared with actual AOD uncertainties are also affecting the climatology related results. However, for a location with a significant $NO_2$ trend, the AOD trend will be also affected. The Rome AOD climatology and trends is a topic of a follow up, in progress study.

Page 11, lines 311-314: A reference to previous studies in Rome including some climatological data and aerosol types would be useful in this context.

As mentioned above, the climatology of Rome is not the main goal of the study. Some relevant references have been included in the text, which is modified as follows:

*"Interestingly, based on Fig. 6, the highest Pandora $NO_2$ retrievals (reddish colors) are not associated with the highest AOD values, indicating that in Rome the high AOD loadings are not strictly associated with high $NO_2$ pollution events. In fact, high AODs are frequently related to long-range transport of elevated layers of desert dust, fires plumes or a combination of both (e.g., Barnaba et al., 2011; Gobbi et al., 2019; Campanelli et al., 2021; Andrés Hernandez et al., 2022). Hence, it might be worth to modify aerosol retrievals for high $NO_2$ in those pollution-related events with low to medium AOD levels. More about AOD and aerosol type climatology for the Rome area can be found in Di Ianni et al., (2018) and in Campanelli et al. (2022). "*

Andrés Hernández, M. D. et al.: Overview: On the transport and transformation of pollutants in the outflow of major population centres – observational data from the EMeRGe European intensive operational period in summer 2017, Atmos. Chem. Phys., 22, 5877–5924, https://doi.org/10.5194/acp-22-5877-2022, 2022.

Barnaba, F., Angelini, F., Curci, G., and Gobbi, G. P.: An important fingerprint of wildfires on the European aerosol load, Atmos. Chem. Phys., 11, 10487–10501, 10.5194/acp-11-10487-2011, 2011.

Campanelli, M., Iannarelli, A.M., Mevi, G., Casadio, S., Diémoz, H., Finardi, S., Dinoi, A., Castelli, E., di Sarra, A., Di Bernardino, A., Casasanta, G., Bassani, C., Siani, A.M., Cacciani, M., Barnaba, F., Di Liberto, L., Argentini, S.: A wide-ranging investigation of the COVID-19 lockdown effects on the atmospheric composition in various Italian urban sites (AER – LOCUS), Urban Climate, Volume 39, 100954, ISSN 2212-0955, https://doi.org/10.1016/j.uclim.2021.100954, 2021.

Campanelli, M., Diémoz, H., Siani, A. M., di Sarra, A., Iannarelli, A. M., Kudo, R., Fasano, G., Casasanta, G., Tofful, L., Cacciani, M., Sanò, P., and Dietrich, S.: Aerosol optical characteristics in the urban area of Rome, Italy, and their impact on the UV index, Atmos. Meas. Tech., 15, 1171–1183, https://doi.org/10.5194/amt-15-1171-2022, 2022.

Di Ianni A, Costabile F, Barnaba F, Di Liberto L, Weinhold K, Wiedensohler A, Struckmeier C, Drewnick F, Gobbi GP.: Black Carbon Aerosol in Rome (Italy): Inference of a Long-Term (2001–2017) Record and Related Trends from AERONET Sun-Photometry Data. Atmosphere. 9(3), 81, https://doi.org/10.3390/atmos9030081, 2018.

Gobbi, G.P., Barnaba, F., Di Liberto, L., Bolignano, A., Lucarelli, F., Nava, S., Perrino, C., Pietrodangelo, A., Basart, S., Costabile, F., Dionisi, D., Rizza, U., Canepari, S., Sozzi, R., Morelli, M., Manigrasso, M.,

Drewnick, F., Struckmeier, C., Poenitz, K., Wille, H.: An inclusive view of Saharan dust advections to Italy and the Central Mediterranean, Atmospheric Environment, 201, 242-256, 10.1016/j.atmosenv.2019.01.002, 2019.

Page 11, line 332: The values of 1.1% and 1.9% included in this line (as well as in the following lines) don't correspond to the values in the table. Are the authors reducing the floating points in the text? The use of these values can cause confusion in the reader.

The authors thank the referee for noticing that. Indeed, we reduce the floating points in the text. Since a precision higher than one decimal point is not necessary for the percentage values, tables have been revised accordingly. The way the numbers are presented in the text has been changed in order to be consistent with the table.

Page 12, lines 350-352: The authors stated that, according to Table 2, SKYNET retrievals are quite similar irrespective of the TROPOMI data used. However, similar results (low difference with the PNG product) were retrieved also in Table 1 for Pandora. Furthermore, mean deviations of AERONET products also displayed very low values…

Indeed, the mean deviations of AERONET products are very low. However, these values decrease by about a half when the monthly TROPOMI $NO_2$ is used. That's the case for all AERONET products (AOD at 380/440nm and AE). One would expect similar behavior for SKYNET retrievals, which is not the case. This comment refers only to the behavior observed using TROPOMI $NO_2$. In the results presented in Table 1 there is not such a clear pattern of decrease when monthly values are used, even for AERONET products.

Page 12, Figs. 8 and 9: Why not merge these two figures into one?

The figures have been merged into one (Fig. 8).

Page 12, line 363: Please define what "modified AOD" is.

The word "modified" has been changed to "$NO_2$-modified" and a reference to Sections 2.4.1 and 2.4.2 has been added.

Page 12, lines 365-370: I find relevant the lack of information (numbers) to quantify these results.

Numbers have been added in the text as follows:

*"The median AOD bias for AERONET is about 0.003 with a maximum of about 0.02 at the peak of the event. The median and maximum AE biases are 0.014 and 0.11, respectively. It can be also noted that in the case of SKYNET both AOD (median value of ~0.008 with a maximum of ~0.03) and AE deviations*

*(median and maximum values of ~0.03 and 0.10, respectively) are a bit higher compared to the respective AERONET deviations of synchronous data."*

Page 14, section 3.6: The authors stated in section 2.2.1 that level 1.5 SSA AERONET data were used in this paper. However, in this section, it is not clear to me what SSA product was used. If I understand well, a mimic of the AERONET product retrieved by GRASP was used as a reference, instead of the AERONET SSA standard product. If so:

- Please correct the information provided in section 2.2.1 including a suitable explanation of GRASP algorithm and products used in this paper.
- Why not use the real product instead a "mimic" product? At least these two SSA should be compared…

Could you please clarify it?

The paragraph about SSA has been removed from section 2.2.1 and a new section (Sect. 2.6) has been included to clearly explain GRASP algorithm and the two approaches selected for this study.

Additionally, in Section 3.6 specific clarification has been added to explain why the GRASP algorithm has been used for the proposed comparisons instead of the AERONET product. The GRASP and AERONET inversion algorithms are fundamentally very similar. GRASP was borne from the heritage of AERONET. However, the different developments of both codes now imply some differences in the provided retrieval products. Thus, to avoid any source of discrepancy that is not introduced purely by the methodology to account $NO_2$, the authors consider that the most appropriate comparison should be done with two identical applications of GRASP, but with different $NO_2$ information.

Comprehensive and meaningful comparisons of the GRASP and AERONET retrievals is a very interesting topic. However, it would need specific investigation, which is out of the scope of this study.

Page 15, Fig. 12: There is no information about the lower panel plot. Is the SSA difference?

The lower panels represent the probability density functions of the differences in SSA at 440 nm between both GRASP approaches in each of the selected stations (Fig. 11 in the revised manuscript). The corresponding explanations and corrected labels have been added in the manuscript.

Page 15, Fig. 12: Y-axis of the upper plot should be SSA and not $NO_2$.

The labels in the axis of this figure represent the name of the applied methodology (Fig. 11 in the revised manuscript). All points shown there correspond to SSA at 440 nm. Corresponding description has been included in the manuscript.

Page 15, Fig. 12: Information about correlation is written in the text in terms of r-squared while in this figure is expressed as correlation coefficient "R" (in capital letters). Please homogenize.

The text has been revised to be consistent with the figure (Fig. 11 in the revised manuscript).

Page 15, line 454: Again, the numbers provided in the text do not correspond to the ones provided in the plot. It is a matter of rounding correctly to the appropriate significant digit. For example: with RMSE values of 0.035 and 0.031 I don't consider it appropriate to conclude that RMSE is < 0.035. The same for R squared.

A correct rounding has been applied to the corresponding numbers in the text.

Page 15, line 452: Why the threshold of 0.9 DU?

The comparisons and text have been updated with a new threshold of 0.7 DU, which corresponds to the average $NO_2$ concentration of the whole data set (0.4) plus 2 times the standard deviation. This change has been made to provide statistical significance to the selected value.

Page 15, line 453-454: The authors stated that a positive bias of 0.02 was found in conditions of high $NO_2$ concentrations. Are they talking about SSA or $NO_2$? From what figure (upper or lower panel) this result comes from? I see in the lower panel an average difference of 0.01 for $NO_2 > 0.9$ (high $NO_2$ conditions) but 0.02 for all conditions. From where did the authors find this result? I feel lost with this section.

The clarifications in the text and the correct labels of the lower panels in Fig. 11 that have been added to the manuscript now clearly show from what data these conclusions have been reached. All data in Fig. 11 correspond to SSA at 440 nm. The probability density functions representation show a consistent average of the difference between both methodologies of 0.03 for values of SSA lower than 0.9 and a mean difference of 0.01 for values of SSA higher than 0.9.

Page 15, lines 455-458: This sentence seems confusing to the reader. Please rephrase. It has also some typos, like the comma after the word "studies".

The corresponding sentence has been revised to improve clarity as follows:

*"Previous studies have found SSA retrieval uncertainties in the range of 0.02-0.03 (Eck et al., 2003; Corr et al., 2009; Jethva et al., 2014; Kazadzis et al., 2016), whereas the correction, when high $NO_2$ is recorded, is usually higher."*

Page 16, line 477: The general result stated here (AOD differences below 0.01 because of this $NO_2$ correction) seems really relevant. In fact, this is the main result a reader is expecting. However, is this general result written somewhere in the text?

Results have been added in the discussion of Section 3.1 as follows:

*"The estimated AOD and AE deviations are below 0.01 and 0.1, respectively, for the majority of observations, i.e., about 96 - 98% of occurrences for both CNR-ISAC and APL-SAP (see also distributions in Fig. 6). The average AOD bias is between 0.002 ± 0.003 and 0.003 ± 0.003 (with the higher values observed at 380nm), while the average AE bias is ~0.02 ± 0.03. Overall, the mean AOD bias is low compared to the estimated uncertainties for the standard AERONET product, i.e., 0.01 - 0.02 (with the higher errors observed in the UV) (Sinyuk et al., 2020). However, the mean AOD bias for the cases of high $NO_2$ levels (> ~0.7 DU) is ~0.011 ± 0.003 at 440 nm and ~0.012 ± 0.003 at 380 nm for APL-SAP and ~0.009 ± 0.003 at 440 nm and ~0.010 ± 0.003 at 380 nm for CNR-ISAC, which is comparable to the AERONET reported uncertainties. The estimated mean bias of AE retrievals for the cases with high $NO_2$ (> ~0.7 DU) is ~0.08 ± 0.04 for both Rome sites. The threshold for $NO_2$ has been selected as the average Pandora $NO_2$ (~0.4) calculated from the whole data set plus two times the standard deviation...*

*... Similarly to AERONET, the derived AOD and AE biases for SKYNET are below 0.01 and 0.1, respectively, for the majority of observations, i.e., about 85% of occurrences for AOD and about 90% for AE (see also distributions in Fig. 7). The overall average AOD bias is ~0.007 ± 0.003, which can be assumed low considering that Nakajima et al. (2020) have estimated a root-mean-square difference (RMSD) of about 0.03 for wavelengths < 500 nm in city areas in AOD comparisons with other networks. However, the mean AOD bias for the cases with high $NO_2$ levels (> ~0.7 DU) is found to be about 0.018 ± 0.003, which is comparable to the RMSD value reported by Nakajima et al. (2020). The overall average AE bias calculated in this study is ~0.05 ± 0.04, whereas the AE bias averaged over the high $NO_2$ cases is about 0.10 ± 0.05."*

---

## Author Comment (AC2)

**Response to anonymous referee #2**

The reviewer comments are given in black, followed by the authors' response in blue. Text copied from the revised manuscript is in blue italic.

Based on a comment from referee #1, Fig. 8 and 9 have been merged and, as a result, the numbering of the following figures has been changed. In addition, the structure of Sect. 2 has been modified in order to include a new subsection with the description and implementation of the GRASP algorithm (Sect. 2.6).

**Reviewers' comments:**

The authors use extensive measurements to investigate the impact of $NO_2$ concentrations on AOD and AE retrievals. This paper contributes to better understanding that considering $NO_2$, which is highly diurnal-variable, is important to improve aerosol properties in the spectral range where $NO_2$ absorption is strong. Since the manuscript is well-written, I think readers may understand your approach and result well. I believe the paper can be published for AMT after addressing the concerned expressed below.

We would like to acknowledge the referee for their helpful and thorough review. We believe that their comments improved the quality of this work.

**Minor Issues and specific comments:**

P4 L104:

In AERONET site information
(https://aeronet.gsfc.nasa.gov/new_web/photo_db_v3/Rome_Tor_Vergata.html),

Rome-Tor Vergata site is located at elevation=130 m but your description is shown as 117 m.

Which one is correct?

117 m is the value given in PGN data files and corresponds to the altitude at ground level. The value of 130 m in AERONET site info refers to the elevation on the roof where the instrument is installed.

P6 L166

Do you use $NO_2$ VCD (vertical column density) or SCD (slant column density) from Pandora product? For clarification, it might be better to mention you use $NO_2$ VCD in Section 2.3.1

We used the vertical column of $NO_2$. The text has been revised accordingly.

P6 L183:

Do Brewer $NO_2$ and Pandora $NO_2$ show good agreement? You need to mention how good quality in your Pandora $NO_2$ measurement since you use Pandora $NO_2$ to correct AERONET and SKYNET operational AOD, AE, and SSA product. More reliable $NO_2$ measurements make your study more meaningful. So, add one or two sentences to show how Pandora $NO_2$ agrees well with $NO_2$ from other instruments.

Results from the comparisons with Brewer and MFDOAS $NO_2$ as well as estimations of the Pandora total $NO_2$ accuracy have been included in the text:

*"Total $NO_2$ data from the Pandora instrument #117 located at APL-SAP have been compared with $NO_2$ observations retrieved by the co-located MkIV Brewer spectrophotometer with serial number #067, revealing a correlation coefficient above 0.96 and a negligible absolute median bias of 0.002 DU (Diémoz et al., 2021). According to Herman et al. (2009), the Pandora direct-sun total $NO_2$ has a clear-sky precision of 0.01 DU in slant column and a nominal estimated accuracy of 0.1 DU in the vertical column. In the same study, a systematic difference of less than 1% was found between the relative slant columns of Pandora and a MultiFunction Differential Optical Absorption Spectroscopy (MFDOAS) instrument."*

P6 L191: The Pandora data -> The Pandora $NO_2$ data

The text has been revised accordingly.

P7 L198: You did not show the absolute $NO_2$ difference. However, I think Pandora $NO_2$ is one of the most essential parts in your method. So, it had better to create this plot in the main or the supplement to show how much absolute difference between Pandora $NO_2$ and climatology OMI. If so, readers will understand your approach better.

The difference between Pandora $NO_2$ and OMI climatology is discussed in the text by presenting both absolute and percentage mean values with standard deviation:

*"AERONET aerosol retrievals seem to significantly underestimate the $NO_2$ abundance over urban and suburban locations with an average absolute difference between the actual Pandora measurements and the estimations from satellite climatology of about 0.15 ± 0.19 DU (61.5 ± 71.5%) and 0.16 ± 0.18 DU (61.5 ± 67.2%) for APL-SAP and CNR-ISAC, respectively."*

We also discuss the range of the derived biases (both absolute and percentage) within which the majority of cases is observed:

*"The majority of PGN-OMI biases lie within 0-0.5 DU corresponding to Pandora values lower than 1 DU. More specifically, 90% of the PGN $NO_2$ data over APL-SAP differ within -0.14 DU (-50%) and 0.44 DU (150%) from OMI climatology, while the respective deviation range between -0.14 and 0.51 DU (-50% − 170%) for CNR-ISAC. However, there are quite a few cases (~9.5% and ~8.8% for APL-SAP and CNR-ISAC, respectively) of higher PGN values (< 2 DU) leading to larger deviations (up to ~1.6 DU for APL-SAP and ~1.5 DU for CNR-ISAC)."*

Thus, we think that the levels of the absolute difference between Pandora NO₂ and OMI climatology, as well as their distribution, are clearly presented by the numbers included in the text.

However, based on this comment and a comment from referee #3, the upper panels of Fig. 4 have been replaced with the time series of Pandora – OMI deviations (see figure below).

[Figure]

P7 P225: Are there any specific reasons to exclude the COVID-19 lockdown period? If so, please mention briefly.

Since the TROPOMI data cover a relatively short period (2018-2021) and Fig. 5 is for visualization purposes only, we excluded the lockdown period in order to prevent the low values observed during that period from affecting the average NO₂ values. A brief explanation has been included in the revised manuscript.

P9 L262: In AERONET (Eck et al., 1999), AE is -> The AERONET AE product (Eck et al., 1999) is

The text has been revised accordingly.

P9 L279: the impact of AOD and AE modified retrievals -> the impact of modified AOD and AE retrievals

The text has been revised accordingly.

P9 L280: to investigate the possible effect on the AOD and AE trends -> to investigate the possible effect of $NO_2$ absorption on the AOD and AE trends

The text has been revised accordingly.

P10 L299: to investigate the impact of AOD and AE modified calculations on the derived temporal trends -> to investigate the impact of modified AOD and AE calculations on the derived temporal trends

The text has been revised accordingly.

P11 L311: Any references? Or is this your finding in this research? Then, plot it to explain or direct the figure you show this. You can show the correlation between $NO_2$ and AOD.

This is a finding from Fig. 6. Reddish colors (indicating high $NO_2$ values) do not correspond to high AOD loadings. A reference to the figure has been added in the text.

P11 L336: Do you have any reason to use SKYNET AE for 400-1020 nm?

You use AERONET AE for 440-870 nm. Then, is it more consistent to use similar wavelength pair like SKYNET AE for 400-870 nm?

The aim of this study was to investigate the impact of $NO_2$ absorption on the standard network products, i.e., the products officially available online. The only AE product available from SKYNET is at wavelengths 400-1020nm. In addition, there is not any AE product from AERONET at a spectral range closer to 400-1020nm than AE at 440-870nm.

P11 L338: You show how modified AOD and AE by considering Pandora $NO_2$ and then show modified AOD and AE by implementing TROPOMI $NO_2$. Reader can ask how Pandora $NO_2$ and TROPOMI $NO_2$ are consistent. It had better to add one or two sentences to show how both $NO_2$ measurements are in good agreement. You can refer previous studies about this.

References of TROPOMI and Pandora total $NO_2$ comparison studies have been included in the manuscript:

*"Despite the improved spatial resolution of TROPOMI, the $NO_2$ corrections using TROPOMI data are expected to be less accurate than those performed with the Pandora product. For example, Lambert et al. (2021) showed a bias between TROPOMI and Pandora total $NO_2$ column ranging from -23% over polluted stations to +4.1% over clean areas with a median bias of -7.1%, in the frame of the standard validation process of TROPOMI Level 2 $NO_2$ products. Other studies have concluded similar results. For example, Zhao et al. (2020) showed a negative bias for the standard TROPOMI total $NO_2$ product in the range 23 - 28% over urban and suburban environments and a positive bias of 8 - 11% at a rural site,*

*while Park et al. (2022) showed 26 - 29% negative bias and $R^2$ within 0.73-0.76 over the Seoul Metropolitan Area in Korea."*

P13 L381: The results -> The results in Table 3

The text has been revised accordingly.

P13 L381-388: The description is the analysis in Table 3. Readers may also be curious about the trend itself. AE trends in CNR-ISAC and APL-SAP shows positive and negative, respectively. Do you have any interpretation for this? Is it because inhomogeneous local emission patterns and photochemical destruction you mentioned in P15 L465? Or during your trend analysis period, were there more frequent transports of dust from Africa and caused it negative AE trend in APL-SAP?

We think that it is not possible to answer to this question without speculating based on the absolute changes of the AE, the limited (for such analysis and interpretation) period, the various sources of AE trend uncertainty and the fact that the two datasets are not directly comparable since they are not synchronous. To elaborate a bit more, AE trends for both stations end up in the same range with the AE retrieval uncertainty based on the AOD uncertainty and also comparable with the standard error of the regression slope.

Based on the above reasons, we tried to avoid to present that analysis as a climatology of the area and just used it as an assessment of the error propagation of $NO_2$ correction to AOD and AE trends.

The positive AE trend in the limited time period addressed in this work is just a short-term effect, not a long-term one. A long-term analysis with the CNR-ISAC data (> 20 years) is in progress and will be presented in a follow-up investigation. We can anticipate that, at that site, there is a clear negative trend of fine-fraction AOD, while coarse-AOD keeps almost constant, and this translates into a decreasing AE.

P13 L402: Font type looks different.

The font type has been corrected.

P14 L432: You used not standard AERONET aerosol retrieval but GRASP algorithm.

If both are the same condition, retrieved SSA from GRASP algorithm is the same with that from standard AERONET retrieval? If not, how much difference of SSA is apparent?

In section 3.6 specific clarification has been added to explain why the GRASP algorithm has been used for the proposed comparisons instead of the AERONET product. The GRASP and AERONET inversion algorithms are fundamentally very similar. GRASP was borne from the heritage of AERONET. However, the different developments of both codes now imply some differences in the provided retrieval products. Thus, to avoid any source of discrepancy that is not introduced purely by the methodology to account

NO₂, the authors consider that the most appropriate comparison should be done with two identical applications of GRASP, but with different NO₂ information.

Comprehensive and meaningful comparisons of the GRASP and AERONET retrievals is a very interesting and necessary study. However, the level of required detail and deepness is totally out of the scope of this study.

Also, when you use SSA from AERONET, there are quality assurance criteria (Mok et al., 2018). In Figure 12, do you plot SSA when AOD > 0.4? SSA when AOD is small shows large error.

The authors agree with the referee in the conditions established by Mok et al. (2018) as well as by Sinyuk et al. (2020). However, these restricted conditions imply an extremely reduced amount of available data that makes impossible comparisons with a proper level of statistical significance. Thus, the authors consider that the trade between the amount of data and the loose of accuracy in the retrieved values results beneficially for the final quality of the comparisons. This methodology has been successfully applied in several publications as for example in Román et al. (2017), Román et al. (2018), Benavent-Oltra et al. (2019) and Herreras et al. (2019).

However, despite these uncertainties, in the methodology proposed here the comparison is made with identical retrieval schemes but with different NO₂ representation. Thus, even if random error is present in the retrieved values of SSA at 440 nm, the error observed here is a systematic bias. This is why the conclusions about the need of a correct representation of this gaseous absorption in AERONET-like retrievals are not affected by possible inconsistencies in the amount of information available under AOD or Solar Zenith Angle conditions.

Román, R., Torres, B., Fuertes, D., Cachorro, V. E., Dubovik, O., Toledano, C., ... & Alados-Arboledas, L. (2017). Remote sensing of lunar aureole with a sky camera: Adding information in the nocturnal retrieval of aerosol properties with GRASP code. Remote Sensing of Environment, 196, 238-252.

Román, R., Benavent-Oltra, J. A., Casquero-Vera, J. A., Lopatin, A., Cazorla, A., Lyamani, H., ... & Alados-Arboledas, L. (2018). Retrieval of aerosol profiles combining sunphotometer and ceilometer measurements in GRASP code. Atmospheric Research, 204, 161-177.

Benavent-Oltra, J. A., Román, R., Casquero-Vera, J. A., Pérez-Ramírez, D., Lyamani, H., Ortiz-Amezcua, P., ... & Alados-Arboledas, L. (2019). Different strategies to retrieve aerosol properties at night-time with the GRASP algorithm. Atmospheric Chemistry and Physics, 19(22), 14149-14171.

Herreras, M., Román, R., Cazorla, A., Toledano, C., Lyamani, H., Torres, B., ... & de Frutos, A. M. (2019). Evaluation of retrieved aerosol extinction profiles using as reference the aerosol optical depth differences between various heights. Atmospheric Research, 230, 104625.

In addition, for SSA calculation, I am wondering you use the consistent surface albedo for SSA retrievals. Incorrect surface albedo makes a systematic bias in SSA retrievals (Mok et al., 2018).

Mok, J., Krotkov, N. A., Torres, O., Jethva, H., Li, Z., Kim, J., Koo, J.-H., Go, S., Irie, H., Labow, G., Eck, T. F., Holben, B. N., Herman, J., Loughman, R. P., Spinei, E., Lee, S. S., Khatri, P., and Campanelli, M.: Comparisons of spectral aerosol single scattering albedo in Seoul, South Korea, Atmos. Meas. Tech., 11, 2295–2311, https://doi.org/10.5194/amt-11-2295-2018, 2018.

Both GRASP retrieval schemes are applied using the BRDF as described in Román et al. (2018), a bi-weekly climatology of MODIS BRDF product over the corresponding AERONET sites.

Lastly, overestimation in AOD lead to the underestimation in SSA. When you compare SSA from GRASP/Standard AERONET with that from GRASP/Pandora NO₂, do you use the same AOD?

For this, in figure 12, you should add a plot of difference of SSA as a function of difference of AOD.

*The authors agree that AOD and SSA tends to be inversely correlated, and actually for the retrievals included in the figure (Fig. 11 in the revised manuscript) there is an underestimation of AOD represented by a MBE of -0.0068 (-4.5%). However, the retrieval is complex and a lot of parameters are part of the fitting. Thus, in order to fit TOD and almucantar the size distribution or particle sphericity can be affected by these different NO₂ conditions, which makes the direct connection between biases in AOD and SSA more complex.*

*The correlation between the differences in both magnitudes can be found below for both stations:*

[Figure]

*As it can be seen for AOD differences of less than 0.005, the differences in SSA remains very close to zero. However, establishing a direct relationship between both magnitudes require a deeper look to all parameters used to model aerosol particles.*

P15 L458 or in conclusion:

You may add one or two sentences about the importance of your research to estimate the effect of NO₂ on the spectral dependence of SSA (i.e., absorption Ångström exponent (AAE)) as a future study.

The following text has been included in the conclusions:

*"In future studies, the effect of NO₂ correction on absorption Ångström exponent (AAE) could be explored. AAE is an aerosol optical property that describes the absorption variation with respect to wavelength and is significantly influenced by particle size, shape, and chemical composition used for aerosol characterization and apportionment studies (e.g., Schuster et al., 2006). Since AAE is a function*

*of spectral AOD and spectral SSA, the NO$_2$ correction for certain AOD wavelengths and SSAs shown in this study is expected to impact the AAE calculations."*

Schuster, G. L., Dubovik, O., and Holben, B. N.: Angstrom exponent and bimodal aerosol size distributions, J. Geophys. Res., 111, D07207, https://doi.org/10.1029/2005JD006328, 2006.

P27 L800 (Table 1)

Why NO$_2$ values in Table 1 is different in different wavelengths? Is this because the number of data you used for 380 and 440 is different? Why don't you use the same number of data at all wavelengths? Since we look at AE which is the relationship of AOD between wavelengths, I think you should match the data for all wavelengths. In case one event has some information at one wavelength is missing, it is caused by some issues like small fraction of cloud is passing etc.

The NO$_2$ quantity available from AERONET is the NO$_2$ optical depth. The NO$_2$ column in DU has been derived from the AERONET optical depth values using appropriate cross sections for each wavelength. Slight differences are observed between different wavelengths in the calculated columns (negligible differences in the third decimal point in DU) due to possible minor biases in the cross section used.

Another reason for the different deviations for different wavelengths presented in Tables 1 and 2 is, indeed, the different number of data. However, these differences in the number of data are not due to sparse events, but they refer to whole periods during which the Cimel instrument would not perform measurements in one wavelength or another for any reason.

Cases with missing measurements for one or another wavelength were excluded from AE calculations.

P41 L890 (Figure 12)

In upper left figure, the number of data shown in the figure is not the same with the legend (N=32). Also, there is no explanations for different color (e.g., green and red dots). It is hard to recognize the dots in the plot. Please modify them with increasing size.

The number of points in the plot is correct. The color is an indicator of the density of points, i.e., colors closer to red indicate higher amount of points close together. A very high density of cases with SSA values > 0.95 is observed and this is why it is difficult to distinguish by eye the total number of points stated in the legend. The size of the points has been increased and explanation for the different colors has been added (Fig. 11 in the revised manuscript).

---

## Author Comment (AC3)

**Response to anonymous referee #3**

The reviewer comments are given in black, followed by the authors' response in blue. Text copied from the revised manuscript is in blue italic.

Based on a comment from referee #1, Fig. 8 and 9 have been merged and, as a result, the numbering of the following figures has been changed. In addition, the structure of Sect. 2 has been modified in order to include a new subsection with the description and implementation of the GRASP algorithm (Sect. 2.6).

**General Comments**

This study aims to evaluate the impact of the $NO_2$ correction for the aerosol retrievals based on ground-based instruments (i.e., AERONET and SKYNET). They utilized multiannual data collected at urban and suburban sites in Rome, Italy. For the $NO_2$ correction, they used ground-based Pandora instruments as well as the TROPOMI data. The $NO_2$-corrected aerosol retrievals are compared with the operational methods to assess their effects. This manuscript analyzed valuable collocated data from the AERONET, Pandora, and SKYNET, and presented various results using the data. However, I do not fully agree with their main conclusions, which insist significance of the $NO_2$ corrections for the AERONET and SKYNET products. In the major part of the results, the effects of $NO_2$ correction seem to be negligible to me, which is the reason why the previous algorithms neglected $NO_2$ effects or utilized climatology. I believe the authors need to demonstrate their conclusions based on the statistical test to assess the impacts of the $NO_2$ corrections on aerosol retrievals. Therefore, I would recommend considering the publication of this manuscript after clarifying the below comments.

We would like to acknowledge the referee for their helpful and thorough review. We believe that their comments improved the quality of this work.

In general, we agree that, in the major part of the results, the effects of $NO_2$ correction seem not to be so significant, which is the reason why the previous algorithms neglected $NO_2$ effects or utilized climatology. However, we think that the average systematic underestimation of AOD found for SKYNET (0.007) cannot be considered negligible. Moreover, according to the findings of this study, significant errors may be introduced over polluted areas for cases with high $NO_2$. Those cases are quite a few for Rome, but the error introduced is comparable to the AOD uncertainties. In addition, there are areas with higher $NO_2$ levels and more frequent events of high $NO_2$ compared to the Rome stations used in this study. For studies that do not deal with averages and use individual days in the analysis, the $NO_2$ correction could be important when intraday $NO_2$ variability is high especially in cities or in episodic $NO_2$ cases.

Statistics and references have been included in the text to support the importance of $NO_2$ correction in the above cases. Also, revisions have been made in parts of the manuscript where the significance of the results may has been excessively or inappropriately overstated.

We answer to each point in detail below.

**Major comments**

Lines 316-317: This overestimation should be quantified by suggesting statistical values in the main script although the values are listed in the tables. The values should be compared with the reported uncertainties of the AERONET (i.e., 0.01 in the visible and NIR and 0.02 in the UV) and SKYNET.

The resulted values of mean deviations from our analysis and reported uncertainties for the two networks have been included in the text as follows:

[revised manuscript text omitted]

Lines 415-418: As this result is one of the main conclusions, the authors should report the statistical significance of differences between original and modified data.

We agree with the reviewer that the average effects of $NO_2$ correction are relatively small, which is why the previous algorithms neglected $NO_2$ effects or utilized $NO_2$ climatology. As already highlighted in the paper, the proposed correction and the consequent improvement are, on average, not statistically significant. This result is explainable considering that the suggested correction depends on the amount of $NO_2$ and that the relative frequency distributions of absolute Pandora-OMI deviation decrease for high $NO_2$ values (lower panel of Fig. 4). Basically, we are focusing on those situations in which the $NO_2$ climatology is not able to represent the real scenario. The present work highlights that when significant discrepancies between climatology and PGN $NO_2$ values are observed, the improvement due to the proposed correction is also statistically significant, i.e., larger than combined instantaneous uncertainties.

To better highlight this result, we decided to update Fig. 10, adding in the lower panels the absolute correction as a function of the corresponding MODIS DB AOD data and the absolute difference between PGN and climatology $NO_2$ data for CNR-ISAC (left panel) and APL-SAP (central panel) sites. This type of plot is not included for ESR data, since it would be identical to the upper right panel of the figure, as $NO_2$ absorption is not accounted in the official SKYNET retrieval chain.

[Figure]

The last part of section 3.5 has been revised as follows:

*"This inter-comparison exercise demonstrated that the proposed correction slightly improves the agreement between MODIS DB AOD data and AERONET and SKYNET AOD products, even if, on average, it is not statistically significant. Nevertheless, as shown in Fig. 10, the improvement becomes significant when the differences between the $NO_2$ values observed by Pandora and the OMI $NO_2$ climatology are also significant (lower panels of Fig. 10). Furthermore, since the proposed correction depends on the amount of $NO_2$, the improvement is more evident in the correspondence of high values of $NO_2$ (upper panels of Fig. 10), typical of highly polluted areas such as the urban area of Rome (APL-SAP). Also, a slight improvement is also achieved in the suburban area of Rome (CNR-ISAC). Finally, in the case of SKYNET AOD products, the systematic overestimation, due to neglected $NO_2$ extinction in the official retrieval chain, is eliminated."*

The caption of Fig. 10 was also changed as follows:

*"Upper row: Absolute correction as a function of the corresponding MODIS DB AOD data and PGN $NO_2$ data for CNR-ISAC (left panel) and APL-SAP (middle and right panels) sites. In the left and middle panels, the inter-comparison was performed using AERONET AOD products, in the right panel SKYNET AOD was used. The color scale represents the PGN $NO_2$ retrieved in correspondence with the AERONET/SKYNET AOD products. The analysis was performed considering a maximum distance between the center of the MODIS DB pixel and the site location of 5 km and Δt_max of ±30 minutes. Lower row: As in the upper row, but the color scale represents the absolute difference between PGN and OMI climatological $NO_2$ data in correspondence with the AERONET AOD products."*

Section 3.6: I believe this section is one of the most meaningful results to me. If the impact of the $NO_2$ corrections on the AOD and trend analysis is not statistically significant, I recommend elaborating on this section (e.g., adding more cases or locations, etc.).

We tried to focus on Rome datasets, as the setup of having two sites in such a close distance, two $NO_2$-retrieving photometers and three AOD-retrieving ones is unique.

In general, we think that we demonstrated that not accounting for $NO_2$ or using $NO_2$ climatologies, which are systematically lower than the actual $NO_2$ measured in real time, introduces a systematic error on AOD retrievals. This error is low and within the AOD reported uncertainties on an average level, but it becomes more significant for a number of cases with relatively high $NO_2$.

This is also the case for SSA. We aimed to demonstrate that different than near real-time measured $NO_2$ could affect SSA retrievals in certain wavelengths. So, inversion algorithms for retrieving properties like SSA need to account for $NO_2$ for "high" $NO_2$ cases, where "high" is defined by the $NO_2$ climatology used.

For all sites globally, the effect would be directly proportional to the difference of the climatological $NO_2$ from the actual $NO_2$ for each specific case/measurement. So another study could probably shed light on how accurate are satellite-based climatologies compared with existing ground-based data. Such a study, which is beyond the scope of our analysis, could probably be then used in order to revise the $NO_2$ inputs in the aerosol retrieval algorithm. Of course, in the case of co-located $NO_2$-retrieving instruments at the same site, the AOD retrieval algorithms could be fed with real-time measured $NO_2$.

Lines 463-464: I don't agree that the difference (i.e., lower than 0.003 in table 1) is "quite significant errors" as the errors are typically smaller than the reported uncertainties of the AERONET and/or SKYNET.

The statement has been revised as follows:

*"However, significant errors could be introduced in the AOD retrievals, especially over urban areas, where $NO_2$ variability can be high and also the occurance of high $NO_2$ events can be more frequent. Such errors may occur only in the cases where $NO_2$ is not taken into account or the used $NO_2$ climatology underestimates such high-$NO_2$ events."*

This statement refers to the significant errors that may be introduced over polluted areas for cases with high $NO_2$. Those cases are quite a few, but the error introduced is comparable to the AOD uncertainties. In addition, there are areas with higher $NO_2$ levels and more frequent events of high $NO_2$ compared to the Rome stations used in this study.

The mean bias derived for the high $NO_2$ cases (> ~0.7 DU) in our study is ~0.011 ± 0.003 at 440 nm and ~0.012 ± 0.003 at 380 nm for APL-SAP and ~0.009 ± 0.003 at 440 nm and ~0.010 ± 0.003 at 380 nm for CNR-ISAC for AERONET AOD and ~0.08 for both Rome sites for AERONET AE. In the case of SKYNET, the mean bias for the cases with high $NO_2$ levels (>~0.7 DU) is ~0.018 and ~0.10 for AOD and AE, respectively. These numbers have been included in the manuscript (in Section 3.1 as well as in the abstract and conclusions).

Lines 477-479: Again, according to table 2, it is lower than 0.0011 for AERONET, and 0.0051 for SKYNET, which is much lower than 0.01. I don't believe it is significant given that the AEORNET uncertainty is higher than 0.01.

The corrections in Table 2 are based on space-borne $NO_2$ data. The purpose for including them is to show the possibility for corrections on a global scale. The underestimation of TROPOMI $NO_2$ compared to Pandora leads to lower and less accurate AOD corrections. However, in the case of high $NO_2$ (> ~0.7 DU) the corrections are not negligible. More specifically, a mean AOD bias of ~0.004 ± 0.001 at 440 nm and ~0.005 ± 0.002 at 380 nm for AERONET APL-SAP and ~0.003 ± 0.001 at both 440 nm and 380 nm for AERONET CNR-ISAC was estimated. The mean bias of AE retrievals is ~0.05 ± 0.04 and ~0.02 ± 0.01 for APL-SAP and CNR-ISAC, respectively. In the case of SKYNET, the average bias is about 0.011 ± 0.002 and ~0.07 ± 0.04 for AOD and AE, respectively. These numbers have been included in the manuscript in Section 3.2.

Table 1, which is based on less uncertain ground-based $NO_2$ measurements, shows a 0.002-0.003 (depending on wavelength) and 0.007 difference on the average for AOD for AERONET and SKYNET, respectively. Especially for SKYNET, we think that an average systematic underestimation of AOD of 0.007 cannot be considered negligible, having also in mind that there are parts of the world with much higher average $NO_2$.

WMO (2005) states that 95% of AOD differences compared with a reference standard should lie within ± (0.005 + 0.01/m) of AOD, where m is the optical air mass. The first term of equation (0.005) represents the maximum tolerance for the uncertainty due to the atmospheric parameters used for the AOD calculation (additional atmospheric trace gas corrections, i.e., Ozone and $NO_2$, and Rayleigh scattering). The second term (0.01/m) describes the calibration-related relative uncertainties (WMO recommends an upper limit for the calibration uncertainty of 1 % (e.g., Cuevas et al., 2019, Kazadzis et al., 2018a)).

Based on the above, we consider the average systematic AOD underestimation found in our study, mainly the 0.007 (Table 1 / using Pandora $NO_2$) and 0.005 (Table 2 / using TROPOMI $NO_2$) for SKYNET, important to be reported here.

The above discussion and references have been added in Section 3.1.

WMO: WMO/GAW Experts Workshop on a Global Surface-Based Network for Long Term Observations of Column Aerosol Optical Properties, GAW Report No. 162, WMO TD No. 1287, available at: https://library.wmo.int/index.php?lvl=notice_display&id=11094, 2005.

Cuevas, E., Romero-Campos, P. M., Kouremeti, N., Kazadzis, S., Räisänen, P., García, R. D., Barreto, A., Guirado-Fuentes, C., Ramos, R., Toledano, C., Almansa, F., and Gröbner, J.: Aerosol optical depth comparison between GAW-PFR and AERONET-Cimel radiometers from long-term (2005–2015) 1 min synchronous measurements, Atmos. Meas. Tech., 12, 4309–4337, https://doi.org/10.5194/amt-12-4309-2019, 2019.

Kazadzis, S., Kouremeti, N., Nyeki, S., Gröbner, J., and Wehrli, C.: The World Optical Depth Research and Calibration Center (WORCC) quality assurance and quality control of GAW-PFR AOD measurements, Geosci. Instrum. Method. Data Syst., 7, 39–53, https://doi.org/10.5194/gi-7-39-2018, 2018a.

Lines 505-506: I don't agree that $NO_2$ absorption is very important for the AE, AOD, and SSA retrievals.

We agree with the referee that this is a very strong statement based on the results presented here.

The text has been revised as follows:

*"In general, the effect of $NO_2$ absorption can be relatively important in the retrievals of aerosol properties, especially AE, AOD and SSA at 440 nm and 380nm, when $NO_2$ is not included in the retrieval algorithms or in cases where $NO_2$ absorption is significantly higher than the $NO_2$ climatology used."*

**Minor comments**

Lines 61-62: Please add references of the SKYNET, GAW-PFR, AERONET regarding the $NO_2$ corrections for the aerosol retrievals.

The following references have been added in the text:

AERONET - Giles, D. M., Sinyuk, A., Sorokin, M. G., Schafer, J. S., Smirnov, A., Slutsker, I., Eck, T. F., Holben, B. N., Lewis, J. R., 645 Campbell, J. R., Welton, E. J., Korkin, S. V., and Lyapustin, A. I.: Advancements in the Aerosol Robotic Network (AERONET) Version 3 database – automated near-real-time quality control algorithm with improved cloud screening for Sun photometer aerosol optical depth (AOD) measurements, Atmos. Meas. Tech., 12, 169–209, https://doi.org/10.5194/amt-12-169-2019, 2019.

GAW-PFR - Kazadzis, S., Kouremeti, N., Nyeki, S., Gröbner, J., and Wehrli, C.: The World Optical Depth Research and Calibration Center (WORCC) quality assurance and quality control of GAW-PFR AOD measurements, Geosci. Instrum. Method. Data Syst., 7, 39–53, https://doi.org/10.5194/gi-7-39-2018, 2018a.

SKYNET - Nakajima T., Campanelli, M., Che, H., Estellés, V., Irie, H., Kim, S.-W., Kim, J., Liu, D., Nishizawa, T., Pandithurai, G., Soni, 740 V. K., Thana, B., Tugjsurn, N.-U., Aoki, K., Go, S., Hashimoto, M., Higurashi, A., Kazadzis, S., Khatri, P., Kouremeti, N., Kudo, R., Marenco, F., Momoi, M., Ningombam, S. S., Ryder, C. L., Uchiyama, A., and Yamazaki, A.: An overview of and issues with sky radiometer technology and SKYNET, AMT, 13, 4195-4218, 2020.

Figure 4: I'm not quite sure if the upper panels of figure 4 are meaningful. I would recommend adding temporal plots of the biases (Pandora - OMI) vs. time over whole measurement periods. I believe that chart can show how the simple assumption of the AERONET can affect the temporal analysis of the AOD over a few years.

The upper panels of Fig. 4 have been replaced with the time series of Pandora – OMI deviations (see the following figure).

[Figure]

Lines 194-197: Underestimation of satellite $NO_2$ retrievals (e.g., OMI, TROPOMI) compared to ground-based retrievals (e.g., MAX-DOAS, Pandora, etc) is quite a well- known phenomenon and it is attributable to the different field of view (FOV). I think it is worth noting that $NO_2$ correction using the Pandora is more accurate than the satellite retrievals since the FOV of the Pandora is similar to that of the AERONET in the main script.

Discussion on the underestimation of satellite $NO_2$ retrievals due to their limited spatial resolution has been added in the manuscript.

In Section 2.3.1:

*"This underestimation of the $NO_2$ levels over urban locations, characterized by strong spatial gradients, can be attributed to the fact that OMI climatology cannot capture the temporal and spatial $NO_2$ variability within an urban context (e.g., Drosoglou et al., 2017; Herman et al., 2019)."*

In Section 3.2:

*"Satellite sensors perform measurements globally and provide information on the air quality even over regions that lack ground-based observations. However, as already mentioned for OMI in Sect. 2.3.1, the spatial resolution of the satellite retrievals is limited by the pixel size… Despite the improved spatial resolution of TROPOMI, the $NO_2$ corrections using TROPOMI data are expected to be less accurate than those performed with the Pandora product. For example, Lambert et al. (2021) showed a bias between TROPOMI and Pandora total $NO_2$ column ranging from -23% over polluted stations to +4.1% over clean areas with a median bias of -7.1%, in the frame of the standard validation process of TROPOMI Level 2*

*$NO_2$ products. Other studies have concluded similar results. For example, Zhao et al. (2020) showed negative bias for the standard TROPOMI total $NO_2$ product in the range 23 - 28% over urban and suburban environments and a positive bias of 8 - 11% at a rural site, while Park et al. (2022) showed 26 - 29% negative bias and $R^2$ within 0.73-0.76 over the Seoul Metropolitan Area in Korea."*

Lines 305-306: Is there any reason for the opposite definition between $\Delta\tau$ and $\Delta\alpha$?

Both $\Delta\tau$ and $\Delta\alpha$ are defined in the calculations as the difference of the standard minus the modified value. The equation in the text was wrong. The manuscript has been revised accordingly.

Lines 311-312: This sentence is not clear to me. Typical "pollution events" do not always accompany high loadings of $NO_2$, which depends on emissions sources and environmental conditions. Also, Figure 4-6 does not directly demonstrate the relationship between the AOD and $NO_2$. Scatter plots between AOD and $NO_2$ might be helpful for this statement.

This is a finding from Fig. 6. Reddish colors (indicating high $NO_2$ values) do not correspond to high AOD loadings. The text has been revised so that it is clearer that we refer to high $NO_2$ episodes and a reference to the figure has been added:

*"Interestingly, based on Fig. 6, the highest Pandora $NO_2$ retrievals (reddish colors) are not associated with the highest AOD values, indicating that in Rome the high AOD loadings are not strictly associated with high $NO_2$ pollution events. In fact, high AODs are frequently related to long-range transport of elevated layers of desert dust, fires plumes or a combination of both (e.g., Barnaba et al., 2011; Gobbi et al., 2019; Campanelli et al., 2021; Andrés Hernandez et al., 2022). Hence, it might be worth to modify aerosol retrievals for high $NO_2$ in those pollution-related events with low to medium AOD levels. More about AOD and aerosol type climatology for the Rome area can be found in Di Ianni et al., (2018) and in Campanelli et al. (2022)."*

Andrés Hernández, M. D. et al.: Overview: On the transport and transformation of pollutants in the outflow of major population centres – observational data from the EMeRGe European intensive operational period in summer 2017, Atmos. Chem. Phys., 22, 5877–5924, https://doi.org/10.5194/acp-22-5877-2022, 2022.

Barnaba, F., Angelini, F., Curci, G., and Gobbi, G. P.: An important fingerprint of wildfires on the European aerosol load, Atmos. Chem. Phys., 11, 10487–10501, 10.5194/acp-11-10487-2011, 2011.

Campanelli, M., Iannarelli, A.M., Mevi, G., Casadio, S., Diémoz, H., Finardi, S., Dinoi, A., Castelli, E., di Sarra, A., Di Bernardino, A., Casasanta, G., Bassani, C., Siani, A.M., Cacciani, M., Barnaba, F., Di Liberto, L., Argentini, S.: A wide-ranging investigation of the COVID-19 lockdown effects on the atmospheric composition in various Italian urban sites (AER – LOCUS), Urban Climate, Volume 39, 100954, ISSN 2212-0955, https://doi.org/10.1016/j.uclim.2021.100954, 2021.

Campanelli, M., Diémoz, H., Siani, A. M., di Sarra, A., Iannarelli, A. M., Kudo, R., Fasano, G., Casasanta, G., Tofful, L., Cacciani, M., Sanò, P., and Dietrich, S.: Aerosol optical characteristics in the urban area of

Rome, Italy, and their impact on the UV index, Atmos. Meas. Tech., 15, 1171–1183, https://doi.org/10.5194/amt-15-1171-2022, 2022.

Di Ianni A, Costabile F, Barnaba F, Di Liberto L, Weinhold K, Wiedensohler A, Struckmeier C, Drewnick F, Gobbi GP.: Black Carbon Aerosol in Rome (Italy): Inference of a Long-Term (2001–2017) Record and Related Trends from AERONET Sun-Photometry Data. Atmosphere. 9(3), 81, https://doi.org/10.3390/atmos9030081, 2018.

Gobbi, G.P., Barnaba, F., Di Liberto, L., Bolignano, A., Lucarelli, F., Nava, S., Perrino, C., Pietrodangelo, A., Basart, S., Costabile, F., Dionisi, D., Rizza, U., Canepari, S., Sozzi, R., Morelli, M., Manigrasso, M., Drewnick, F., Struckmeier, C., Poenitz, K., Wille, H.: An inclusive view of Saharan dust advections to Italy and the Central Mediterranean, Atmospheric Environment, 201, 242-256, 10.1016/j.atmosenv.2019.01.002, 2019.

Lines 342-343: As spatiotemporal variabilities of the $NO_2$ are significantly high, the authors should state the spatial and temporal window of this collocation.

This information has been included in the text as follows:

*"Based on the current satellite footprint (5.5 km × 3.5 km), a radius of 5 km around each ground-based station was selected for the spatial co-location. The TROPOMI $NO_2$ data were time-interpolated to AERONET and SKYNET measurements."*

Line 403: font of "Wei et al., 2019" needs to be "times new roman"?

The font type has been corrected.

Lines 407-408: Which data were used to calculate the $NO_2$-modified AERONET? (Pandora or TROPOMI?)

The Pandora data were used. This is now mentioned in the text as follows:

*"The $NO_2$-modified AERONET and SKYNET AOD at 470 nm were also computed with the same approach and the AOD and AE retrievals that have been modified using the Pandora $NO_2$ data."*